# Epigenetic Regulation in Exposome-Induced Tumorigenesis: Emerging Roles of ncRNAs

**DOI:** 10.3390/biom12040513

**Published:** 2022-03-28

**Authors:** Miguel Ángel Olmedo-Suárez, Ivonne Ramírez-Díaz, Andrea Pérez-González, Alejandro Molina-Herrera, Miguel Ángel Coral-García, Sagrario Lobato, Pouya Sarvari, Guillermo Barreto, Karla Rubio

**Affiliations:** 1International Laboratory EPIGEN, Consejo de Ciencia y Tecnología del Estado de Puebla (CONCYTEP), Puebla 72160, Mexico; miguel.olmedo@usalud.edu.mx (M.Á.O.-S.); ivonne.ramirez@usalud.edu.mx (I.R.-D.); andrea.perez@usalud.edu.mx (A.P.-G.); alejandro.molina@usalud.edu.mx (A.M.-H.); miguel.coral@usalud.edu.mx (M.Á.C.-G.); sagrario.lobato@usalud.edu.mx (S.L.); pouyasarvari2008@gmail.com (P.S.); guillermo.barreto@univ-lorraine.fr (G.B.); 2Licenciatura en Médico Cirujano, Universidad de la Salud del Estado de Puebla (USEP), Puebla 72000, Mexico; 3Facultad de Biotecnología, Campus Puebla, Universidad Popular Autónoma del Estado de Puebla (UPAEP), Puebla 72410, Mexico; 4Decanato de Ciencias de la Salud, Campus Puebla, Universidad Popular Autónoma del Estado de Puebla (UPAEP), Puebla 72410, Mexico; 5Laboratoire IMoPA, CNRS, Université de Lorraine, UMR 73635 Nancy, France; 6Lung Cancer Epigenetic, Max-Planck-Institute for Heart and Lung Research, 61231 Bad Nauheim, Germany

**Keywords:** epigenetic reprogramming, environment-related toxicants, tumorigenesis, biomarkers, noncoding RNAs, exposome

## Abstract

Environmental factors, including pollutants and lifestyle, constitute a significant role in severe, chronic pathologies with an essential societal, economic burden. The measurement of all environmental exposures and assessing their correlation with effects on individual health is defined as the exposome, which interacts with our unique characteristics such as genetics, physiology, and epigenetics. Epigenetics investigates modifications in the expression of genes that do not depend on the underlying DNA sequence. Some studies have confirmed that environmental factors may promote disease in individuals or subsequent progeny through epigenetic alterations. Variations in the epigenetic machinery cause a spectrum of different disorders since these mechanisms are more sensitive to the environment than the genome, due to the inherent reversible nature of the epigenetic landscape. Several epigenetic mechanisms, including modifications in DNA (e.g., methylation), histones, and noncoding RNAs can change genome expression under the exogenous influence. Notably, the role of long noncoding RNAs in epigenetic processes has not been well explored in the context of exposome-induced tumorigenesis. In the present review, our scope is to provide relevant evidence indicating that epigenetic alterations mediate those detrimental effects caused by exposure to environmental toxicants, focusing mainly on a multi-step regulation by diverse noncoding RNAs subtypes.

## 1. Introduction

The central dogma in genetics is that information in our cells flows only in one direction, from DNA to RNA, then to proteins. It was an absolute dogma that has now been essentially debunked due to the role of the environment in the modulation of gene expression [1]. Environmental factors, including pollutants and lifestyle, constitute a significant role in severe, chronic pathologies with social and economic consequences [2].

The measurement of all environmental exposures and assessing their correlation with effects on individual health is defined as the exposome [3]. An individual’s exposome begins before birth and includes insults from environmental and occupational sources. In fact, the exposome interacts with our unique characteristics such as genetics, physiology, and epigenetics. The “environmental exposure” is a complex construct that encompasses exposures either physical, chemical, biological, or societal [4]. Historically, Christopher Paul Wild defined the exposome in 2005 as the totality of an individual’s exposure experience from conception until death and its impact on chronic diseases [5]. As a concept, it was put forward to stress the necessity of appropriate tools development for exposure assessment when applied to the study of human disease’s etiology [6].

The adaptation of cells to environmental factors causing stress relies on a wide range of tightly controlled regulatory mechanisms. The epigenetic landscape represents the platform where multiple environmental factors interact with the complex genetic milieu, resulting in alterations in the expression gene that shape many aspects of health and disease [7]. Changes in the organization and the structure of chromatin structure are associated with the transcriptional response to stress caused by the environment, and in some cases, can impart the memory of stress exposure to subsequent generations through mechanisms of epigenetic inheritance [8].

Epigenetics investigates modifications in the expression of genes that do not depend on the underlying DNA sequence. Some studies have confirmed that environmental factors, such as toxicants, may promote a phenotype or a disease in an individual or even in the subsequent progeny through epigenetic alterations [9,10]. Several epigenetic mechanisms, including modifications in DNA (e.g., methylation), histones, and non-protein coding RNAs (ncRNAs) can change genome expression under the exogenous influence [11]. Notably, the role of long noncoding RNAs (lncRNAs) in epigenetic processes has recently been highlighted in more detail [12]. The latent interest in epigenetics has resulted in breakthroughs in seminal concepts in diseases ranging from autoimmune conditions to cancer, congenital diseases, mental retardation, endocrine diseases, pediatric diseases, neuropsychiatric disorders, and many others [1,13].

Discrepancies in the homeostatic functions of the epigenetic machinery cause a range of different disorders since these mechanisms (“epigenome”) are more sensitive to the environmental status (lack of nutrient consumption, physicochemical exposures, and psychological stress) than the genome, especially during early development due to the inherently dynamic nature of the epigenetic landscape [14]. Besides the factors above, exposure to pollutants such as arsenic, nickel, cadmium, mercury, benzene, dioxin, bisphenol A, and diethylstilbestrol can alter the epigenetic regulation of the genome. These changes can cause abnormal gene expression, leading to various cancer types and other diseases [15]. Here, we review current evidence indicating that epigenetic alterations mediate those detrimental effects caused by exposure to environmental toxicants, focusing mainly on a direct regulation by a diversity of ncRNAs subtypes (Figure 1).

## 2. ncRNA Subtypes, Biogenesis, and Turnover

### 2.1. miRNAs

MicroRNAs (miRNAs) are the smallest ncRNAs, 20–25 nucleotides long, and do not encode proteins [16]. They bind in a complementary manner in the 3′ untranslated region (3′ UTR) of messenger RNAs (mRNAs) and mostly target them for degradation and blocking of their translation [17]. A miRNA can bind numerous mRNAs to inhibit their translation. Therefore, miRNAs are considered crucial post-transcriptional regulators of gene expression [16]. However, under certain cellular contexts (for instance, cell cycle arrest) they are recruited with AGO2 and FXR1 to specific loci with AU-rich elements (AREs) to ultimately activate translation [18,19,20]. Interestingly, recent evidence has called into question that the function of miRNAs is exclusively in the cytoplasm as a post-transcriptional mechanism targeting only mRNAs. In contrast, nuclear miRNAs potentially act during transcriptional silencing of bi-directionally expressed genes involving the formation of miRNA-ncRNA duplexes and nucleolus organization [21,22].

Genes coding for miRNAs are transcribed by RNA polymerase II (POLII) in the nucleus, where the primary miRNAs (pri-miRNAs) are capped, spliced, and polyadenylated. One miRNA, or clusters of two or more miRNAs, are produced from a primary transcript [16]. The double-stranded RNAse III DROSHA, inside a nuclear protein complex called Microprocessor, exerts the important role of cleaving the immature long pri-miRNAs. One essential cofactor that supports the activity of this protein complex is the double-stranded RNA (dsRNA)-binding protein DiGeorge syndrome critical region 8, DGCR8. Two RNase III domains inside DROSHA are necessary for the cleavage of one strand of the dsRNA to release a hairpin-shaped precursor miRNAs (pre-miRNAs) of approximately 60–70-nucleotides long [17]. Next, the export of the pre-miRNAs to the cytoplasm is mediated by exportin 5 (XPO5), and these are then cleaved by DICER1, an RNase III enzyme, originating from dsRNA to produce the mature miRNA duplex with 2-nucleotide 3′ overhangs of 22 nucleotides. DICER1 associates with transactivation-responsive RNA-binding protein (TRBP), which attaches dsRNA. TRBP physically bridges DICER1 with the Argonaute proteins (AGO1, AGO2, AGO3, or AGO4) and with members of the GW182 protein family during the assembly of the miRNA-induced silencing complex (miRISC). The silencing domain within GW proteins interacts with the deadenylase complex, which shortens the miRNA poly(A) tail. The mRNA-decapping enzyme 1 (DCP1)–DCP2 complex removes the 7-methylguanylate (m7G) cap. Consequently, the unprotected 5′ end is degraded by 5′–3′ exoribonuclease 1 (XRN1). In this manner, a specific mRNA is still stable at early stages, but its translation is inhibited, whereas, at later stages, mRNAs with short poly(A) tails are degraded [17]. Recent evidence showed that the increased production of miRNAs is proportional to the concentration of target mRNAs. Increased processivity of AGO2-associated DICER in the presence of target mRNA also contributes to the higher biogenesis of mature miRNAs [23].

Several classes of non-canonical, Drosha-independent or Dicer-independent miRNAs have been identified. One example of miRNAs that are Microprocessor-independent is related to “mirtrons”, which originate from spliced introns that function as pre-miRNAs and can be immediately exported to the cytoplasm for further processing by DICER [16]. Potential miRNAs can also originate from small nucleolar RNAs (snoRNAs) and transfer RNA (tRNA) fragments, and some of them can even be loaded into the RISC complex. Another Microprocessor-independent miRNA type is originated from the 5′-end of POLII-transcribed genes. Early transcription termination liberates the hairpins and serves as a DICER substrate. These miRNA precursors have a 5′7-methylguanylate (m7G) cap, which is not removed by the Microprocessor and facilitates its nuclear export by the cap-binding complex–exportin 1 (EXP1) pathway [16]. In particular, the existence of DICER-independent miRNAs arose from the detailed analysis of AGO2 activity, an RNase H-like endonuclease in *AGO2*-knockout mice [24] and zebrafish [25]. In other studies, focused on the precursor of the erythrocyte-specific *miR-451*, it was demonstrated that sequences that are too short to be processed by DICER produce abundant miRNA loads. Thus, *miR-451* is bound by AGO2 and cleaved within its stem for mature and functional miRNA formation. This crucial process is conserved during erythrocyte maturation [24,25].

### 2.2. lncRNAs

Long noncoding RNAs (lncRNAs) lack protein-coding properties and extend more than 200 nucleotides. Although the proportion of functional lncRNAs is still undetermined and investigation is required to support evidence of functionality of the majority of lncRNAs, it is well documented that a growing number of lncRNAs have determinant roles in different mechanisms of gene regulation [26]. These large molecules of RNA can be spliced and polyadenylated, although some of them bind to ribosomes [27]. LncRNAs were originally described to have equivalent chromatin features to protein-coding genes [28]. However, recent work has highlighted the differences in the abundance of precise histone marks and splicing efficiency between lncRNAs and coding RNAs [29], as well as additional specifications among lncRNA subsets that differ in their chromatin landscapes [30].

LncRNAs are fundamental to different mechanisms such as gene imprinting, cells, and tissue differentiation, antiviral response, and the development of cancers. Among the broad mechanisms of action of transcriptional regulation by lncRNAs, those considered most relevant and universal is the interaction with chromatin-modifying complexes, obstruction of the transcriptional machinery, maintenance of the structure of nuclear speckles, regulation of splicing, regulation of mRNA degradation, modulation of protein translation and stability, and acting as molecular sponges for miRNAs [31]. However, while lncRNAs can be dysregulated in various human diseases known to include environmental factors as etiology, it is unclear how specific the interactions between lncRNA and mediators of the cellular response to environmental exposures could be. Most of the available data are derived from cell studies, and no data generated from population-based studies have been published [32]. In addition, as highly specific as lncRNAs are, depending on the target tissue, they represent potential diagnostic markers and, in consequence, therapeutic targets in cancer biology. Developing RNA-targeting therapeutics means a tremendous opportunity for modulating lncRNAs endogenous activity. Several preclinical approaches have been developed in this context, and others are currently under development by several pharmaceutical companies [33].

### 2.3. Other ncRNA Biotypes (siRNA, piRNAs, snoRNAs)

Although miRNAs are by far the most studied class of small-ncRNA biotypes in many fields, including cancer, environmental toxicology, and risk assessment, research of other RNA biotypes such as small nucleolar RNAs (snoRNAs), short interfering RNAs (siRNAs), and PIWI-interacting RNAs (piRNAs) may lead to similar breakthroughs based on their distinct cellular functions. First, siRNAs are 21- and 22-nucleotides RNAs generated by ribonuclease III (RNase III) and cleaved from longer double-stranded RNAs (dsRNAs) involved in post-transcriptional gene silencing in mammals and plants. Silencing is initiated by dsRNA that is homologous in sequence to the silenced gene [12]. Similar to miRNAs, siRNAs depend on DICER enzymes to excise them from their precursors [34] and AGO proteins to support their silencing-effector functions [35]. Generally, siRNA assembles into functional siRISC; one strand is separated and degraded. Then, siRNA-containing siRISC binds to AGO2 protein. This complex finally recognizes its targets by base pairing, and silencing occurs through one of several molecular mechanisms [36]. PIWI-interacting RNAs (piRNAs) are 26–32 nucleotides in length and bind to the PIWI protein family; they are abundant in spermatogenic cells, responsible for stem cell self-renewal, and involved in transposon silencing. piRNAs have been found to act in somatic cells and are crucial in guiding epigenetic regulation [37]. The biogenesis of piRNAs, initially described in *Drosophila* spp. and in *C. elegans*, begins with the transcription of a long, single-stranded precursor derived from a coding- gene transcript, transposons, tRNA, ribosomal RNA (rRNA), or intergenic loci by RNA POLII. Furthermore, a DICER/DROSHA-independent manner makes their processing. The precursor is exported from the nucleus into yeast bodies and further processed into smaller segments in yeast. piRNAs then form a complex with PIWI proteins assisted by shutdown (SHU) and heat shock protein 90 (HSP90). Their mature form is completed when the enzyme HENMT1 adds a methyl group to the 2′ carbon of the ribose on the 3′ end of the transcript. It can reenter the nucleus to induce transposon silencing and epigenetic regulation [38]. The abnormal expression of piRNA is associated with various cancers such as gastric, breast, renal, colorectal, and lung cancer [39].

Small nucleolar RNAs (snoRNAs) are short (60–300 nucleotides long) crucial biomolecules that participate importantly in ribosome biogenesis. They also have a role in the chemical modifications of some RNA biotypes, such as rRNAs, tRNAs, and small nuclear RNAs. There are two classes according to their conserved sequence elements: the C/D box snoRNAs (hairpins containing a sizeable internal loop, bounded by the C/C′ box and the Box D/D’ motifs), and the H/ACA box snoRNAs (two stem-loop structures separated by a conserved single-stranded motif with a consensus sequence of ANANNA) that are predicted to direct site-specific pseudo-uridylation and 2′-O-methylation of rRNA, to guide pre-rRNA processing and to act as molecular chaperones [40]. snoRNAs are derived from either mono- or polycistronic transcription units. In contrast, most of them are encoded within introns of pre-mRNAs and are transcribed mainly by RNA POLII. However, RNA POLIII and their final maturation require exonucleolytic trimming via two pathways: a splicing-dependent and splicing-independent pathway [41]. Splicing of the pre-mRNA results in a debranched precursor further processed by exonucleases to develop a mature snoRNA [42]. Because of their essential roles in biological processes, dysregulations in their expression can actively contribute to the promotion of carcinogenesis in the lung [43] and breast tissue [44].

## 3. Epigenetic Alterations Induced by Toxicants

Environmental toxicants are widespread worldwide and include various chemicals, from combustion products to contaminating trace metals and residual organic compounds used in daily life [45]. Those considered pollutants or chemical agents are chemical elements or compounds whose state and physicochemical characteristics allow them to encounter individuals. Their main routes of entry into the body are respiratory, dermal, and digestive [46]. Exposure to varying levels and types of these contaminants is linked to a substantial harmful impact on human health. For this reason, the environmental health and toxicology fields have recently focused on improving the understanding of how these contaminants impact biological processes so they can be prevented [45]. The following sections introduce a list of the main toxins that act at the epigenetic level, according to the four primary sources of exposure to chemical agents (Figure 2 and Appendix A).

### 3.1. Toxics in the Air

According to the United States National Ambient Air Quality Standard (NAAQS), there are six principal air pollutants: nitrogen dioxide (NO_2_), carbonic oxide (also known as carbon monoxide, CO), ozone (O_3_), sulfurous acid anhydride (also known as sulfur dioxide, SO_2_), lead (Pb), and particulate matter classified as a particulate matter of ≤10 μm (PM10) and ≤2.5 μm (PM2.5) [47,48]. Air pollution has numerous harmful effects and contributes to cardiovascular and lung diseases, including asthma, chronic obstructive pulmonary disease (COPD), and metabolic disorders. Air pollution can induce changes in DNA methylation [49], histone modifications [50], as well as regulation by ncRNAs [51].

#### 3.1.1. Ozone

O_3_ is an environmental toxicant caused by photochemical reactions in automobile exhaust gases [49,50,51,52,53,54]. Respiratory exposure to ozone causes inflammation and injury to the respiratory tract. A recent study investigated the impact of atmospheric pollutants on the murine lung. The results show that the ozone-induced damage increases the markers of inflammation and the number of EV-sized particles in the BALF in murine lungs. Moreover, high-throughput small RNA sequencing identified the altered expression of several microRNAs in pulmonary extracellular vesicles, including *miR**-22**-3p* [55].

#### 3.1.2. Carbon Monoxide

Carbon monoxide (CO) competes with oxygen and alters the hemoglobin dissociation curve. Once it enters into the circulation, it binds to the Heme group of hemoglobin, displacing the oxygen and forming a hematic complex called carboxyhemoglobin, which hinders oxygen transport to cells and tissues, causing generalized cellular hypoxia. Hypoxia is known to affect cells with higher oxygen demand, particularly myocytes and neurons, which might have been linked to a higher risk of cardiovascular and neurodegenerative diseases [56,57]. Recently, novel epigenetic signatures were identified to be associated with the uptake of carbon monoxide (DLCO) per alveolar volume (VA) (DLCO/VA), using the single-breath technique in 2674 individuals. Two CpG sites (cg05575921 and cg05951221) significantly associated with CO uptake were identified. Furthermore, they found a positive association between hypomethylation of the *AHRR* gene (cg05575921) and the expression of *EXOC3* in whole blood, suggesting that *EXOC3* is an excellent candidate through which smoking-induced *AHRR* hypomethylation could affect the exchange of pulmonary gases [58].

#### 3.1.3. Lead

Lead (Pb) is a xenobiotic metal and toxic pollutant. Lead-contamination primary sources are associated with the inhalation of contaminated air and ingestion of contaminated food [59]. Pb affects the nervous system and is associated with cancer and neurodegenerative disorders [59,60]. A group of Caucasian male employees with higher lead concentration in their blood exhibited complete CpG methylation, including the *CDKN2A* promoter, whereas the group with lower lead concentration in their blood showed partial methylation. Therefore, results suggested that DNA methylation could be involved in the mechanism by which it induces neurotoxicity [61]. However, additional studies are required to determine the specific effect in the DNA methylation profile in neuronal cells using in vivo and in vitro experiments.

#### 3.1.4. Sulfur Dioxide

Sulfur dioxide (SO_2_) is a colorless gas that condenses at −10 °C and solidifies at −72 °C. It is soluble in water and organic solvents. During its oxidation process in the atmosphere, it forms sulfates that can be transported in PM10 and, in the presence of humidity, form acids that are components of PM2.5 [61,62]. One study reported that the co-exposure of PM2.5 and SO_2_ at low doses leads to neuronal apoptosis, reduction in the postsynaptic density, and synergistic neurodegeneration by the activity of *miR-337-5p* [63].

#### 3.1.5. Nitrogen Dioxide

Nitrogen dioxide (NO_2_) is a gas that generates inflammation bronchial hyperresponsiveness and causes an imbalance in the Th1/Th2 immune response. Furthermore, prenatal exposure to NO2 is also associated with respiratory infections and asthma development during childhood [64,65]. A longitudinal panel study was conducted among 40 university students with four repeated measurements in Shanghai, finding that short-term exposure to NO_2_ was associated with hypomethylation of *NOS2A*, hypermethylation of *ARG2*, respiratory inflammation, as well as impaired lung function [66].

#### 3.1.6. Particulate Matter

Considering all atmospheric pollutants, particulate matter (PM) is considered a profound health problem because they are made up of a complex combination of elemental and organic carbon, nitrates, volatile organic compounds, sulfates, polycyclic aromatic hydrocarbons, metals, and biological components [67]. PM is classified based on its diameter, including PM10 and PM2.5 [68]. Fine particulate matter enters the lower respiratory tract due to its low diameter, while 10 µm-particles remain in the upper respiratory tract. Thus, most epidemiological reports indicate that PM2.5 is the most harmful fraction for public health [67,69]. In general, PM comprises heavy metals and polycyclic aromatic hydrocarbons, which can induce modifications in DNA methylation, histone modifications, and, importantly, ncRNA expression. Specifically, several studies have revealed the effects of PM2.5 on the deregulation of miRNAs such as *miR-4516* and *miR-32*, whose upregulation is observed in lung cancer cells. Moreover, the downregulation of *RPL37*, a target of *miR-4516*, enhanced the expression of LC3B, a critical hallmark of autophagy [70]. Meanwhile, *miR-32* might function as a tumor suppressor to repress EMT [71]. Conversely, PM2.5 affects the expression of *miR-194-3p* and *miR-16* in human bronchial epithelial cells (CSE-HBEpiCs) treated with cigarette smoke extracts, as well as in human hepatocellular carcinoma (HCC) cells. *miR-194-3p* inhibition, in turn, enhances apoptosis [72] while *miR-16* deregulation could enhance metastasis and EMT features [73]. More recently, studies demonstrated that PM2.5 also transform normal cells by lncRNA alteration. For example, *MEG3* (maternally expressed gene 3) over-expression promotes autophagy and apoptosis by increasing TP53 (a tumor suppressor) in HBE cells after treatment with arterial traffic ambient PM2.5 (TAPM2.5) and wood smoke PM2.5 (WSPM2.5) [74]. *MEG3* is also upregulated in CSE-treated 16HBE cells and COPD tissues, leading to cell proliferation impairment by downregulation of *miR-218* [75]. Another upregulated lncRNA by PM2.5 is *LOC101927514*, which binds to STAT3 to raise an inflammatory state in 16HBE cells [76].

##### Organic and Elemental Carbon

Organic carbon, or secondary organic aerosol, is the main component of PM2.5. It represents between 20–80% of the total mass of these particles and has various chemical forms. Carbonaceous aerosols can be released from multiple sources, including mobile devices, biomass burning, meat cooking, and fossil fuel burning, and are also believed to be generated through the gas phase oxidation of precursors in the atmosphere [77,78]. Elemental carbon in the atmosphere is a dark-colored, low-volatile material that does not evolve below 700 °C. Elemental carbon is also an essential component of PM2.5 and is emitted into the atmosphere by various combustion processes, such as fossil fuels and biomass burning. This type of carbon influences the regional and global climate by directly absorbing solar radiation, changing cloud precipitation, acting as ice nuclei or cloud condensation through coating with soluble species, improving the fusion of the snow and ice cap when deposited on these surfaces [79,80]. During phagocytosis and metabolism, elemental carbon and organic carbon have produced highly reactive oxygen species (ROS) concentrations. ROS causes acute and chronic inflammatory diseases, including cardiovascular disease, asthma, and cancer. Furthermore, many lines of evidence demonstrate that ROS can induce epigenetic changes such as DNA methylation or microRNA differential expression [81]. In addition, ROS induced by PM2.5 exposure may promote the expression of some lncRNAs such as *loc146880* [82], which expression was positively correlated with *RCC2* [83] that encodes the protein RCC2/TD-60, required for cell cycle progression [84]. Furthermore, it was shown that RCC2 can bind and stimulate the effect of the lncRNA *LCPAT1* (lung cancer progression-association transcript 1) on cell autophagy and EMT after exposure to PM2.5 and CSE (cigarette smoke extracts) in lung cancer cells [85]. Therefore, high *loc146880* and *LCPAT1* levels might be associated with lung cancer development [82,83,85].

##### Sulfates

Sulfates (SO_4_) are mainly secondary components that originate from the oxidation of SO_2_, although they may be present as a primary component derived from sea salt or mineral matter, such as gypsum. Sulfate is closely related to the release of CO_2_ into the atmosphere [86]. In the Normative Study of Aging, 1406 blood samples from 706 elderly participants were analyzed for DNA methylation changes in sulfate-associated repetitive elements, a prolonged exposure to SO_4_ particles showed significant association with hypomethylation of the long-interspersed nucleotide element-1 (LINE-1) retrotransposon and the short interspersed nuclear element (SINE) Alu [87].

##### Polycyclic Aromatic Hydrocarbons

Polycyclic aromatic hydrocarbons (PAH) are mainly present in the organic component (Po) of PM2.5 [88] as by-products of combustion and enter the human body from various sources, including inhaling gasoline, diesel-fueled engines, coal, coke, oil burners, eating grilled and smoked meats, and cigarette smoking [63]. Reports have shown PAHs as etiological agents of several types of cancer including colon, pancreas, breast, lung, and prostate. However, the biological pathways by which contaminants cause adverse health effects are largely unknown [89]. Moreover, exposure to PAHs has been investigated to alter glutathione, DNA, and RNA (hydroxy) methylation levels, the formation of DNA PAH-adducts [89], as well as the expression of ncRNAs. For example, some breast tumor-related genes such as *EPHB2* (Ephrin type-B receptor 2) and *LONP1* (Lon Peptidase 1) present altered methylation under PAH and NO2 exposures [68]. On the other hand, chronic Po exposure in lung adenocarcinoma cells enhanced their cancer stem cell properties through a long noncoding *loc107985872*-notch1 signaling pathway which could be recapitulated in vivo [90,91]. Similarly, *MALAT1* could interact with *miR-204* which results in increased *ZEB1* (an EMT-related transcription factor) function which in turn enhances the EMT and malignant transformation in lung bronchial epithelial cells after Po exposure [92].

#### 3.1.7. Indoor Air Pollutants

There are five types of sources of pollution in an ordinary home. The first source to be recognized was the combustion of fuels in heating and cooking food. The most used gases as fuel are natural gas (methane) and liquefied gas (propane-butane), which mainly produce NO_2_ and CO (described previously in Section 3.1.5 and Section 3.1.2, respectively). If wood is used for heating in chimneys or cooking (this is the case in many countries in the world), then PM 2.5 and PM 10 are added and a series of PAHs (described in Section 3.1 and Section 6 and Polyciclic Aromatic Hydrocarbons, respectively).

The second source of internal contamination is the result of natural and synthetic materials used in carpets, foam insulation, interior decoration papers, and furniture. The glues used in chipboard, for example, produce formaldehyde. Latex rugs are a source of phenyl cyclohexane emissions. Asbestos used in building materials for its heat resistance properties can cause the emission of asbestos fibers indoors if they are not adequately sealed. In offices, some types of photocopiers and computer printers are a source of toxic organic substances such as toluene. Indeed, the air in many modern buildings is particularly polluted due to the combination of office equipment, synthetic carpets, and poor ventilation [93,94].

The third possible source of internal contamination is the leakage of toxic gases through the ground of the house or from the sewage services due to potential contamination into the pipes. In the United States and China, the largest source of gas emissions though the ground is the radioactive gas radon [95,96]. Many of the commercial products used domestically, such as furniture cleaners, glues, cleaning agents, cosmetic deodorizers, pesticides, and solvents used at home, contribute to the toxicity of the indoor ambient air. These products are the fourth source of production of internal contamination. The fifth source of pollution is cigarette smoke that emits PM. Not only is a pollutant toxic by itself, but it also increases the risk of diseases from other toxic compounds present in indoor environments [97,98].

##### Formaldehyde

Using proteomic analysis, previous groups have identified that formaldehyde can cause epigenetic changes in human cells that lead to the phosphorylation of histone H3. In addition, cells from the pulmonary tract exposed to this aldehyde in the gaseous state displayed 89 miRNAs significantly downregulated. Those alterations were potentially implicated in signal transduction mechanisms associated with the regulation of the endocrine system, inflammatory response, and cancer, indicating that formaldehyde exposure can change the expression profile of miRNAs, making them potential pathogenic drivers of formaldehyde-induced diseases. The authors suggested that this dysregulation could cause the onset and progression of heart and cardiovascular diseases by modulating the expression levels of ncRNAs [99], as it may occur in congenital heart disease (CHD), where several circRNAs may be dysregulated as shown in fetal heart rat samples exposed to formaldehyde compared to the control group [100].

##### Asbestos

Asbestos is a natural mineral consisting of extremely fine fibers that can become trapped in the lung upon inhalation. Genomic DNA mutations and gene rearrangements that cause lung cancer are extensively reported, with biomarkers and targeted therapies already in clinical use [101]. However, our current understanding of how asbestos fibers trapped in the lungs cause epigenetic changes and lung cancer is incomplete. Redox reactions on the surface of fibers have been shown to generate ROS that in turn damage DNA, causing genetic and epigenetic alterations that reduce the activity of tumor suppressor genes. Asbestos-associated lung cancer exhibits less methylation variability than lung cancers in general, and to a substantial extent, the variability is restricted to promoter regions [102]. Additionally, asbestos exposure is highly related to malignant pleural mesothelioma (MPM) development [103], in which various markers of DNA methylation, ncRNAs, and histone modifications have been proposed as diagnostic markers involved in the progression of MPM [104]. In this sense, *lncRNA-RP1* and *miR-2053* were part of a four-RNA signature in serum, in combination with the DNA damage regulated autophagy modulator 1 (*DRAM1*) and arylsulfatase A (*ARSA*) exclusively in MPM patients [105]. Furthermore, *miR-16* (inhibitor of cell proliferation and migration), *miR-17* (potential tumor suppressor), *miR-126* (inhibitor of VEGF activity), and *miR-486* (antifibrotic) were downregulated in MPM samples. Together, these miRNAs can represent the basis of the mechanism involved in MPM progression [106]. Since GAS5 (Growth arrest-specific transcript 5) might act as an oncogene released by the tumors and upregulated in serum of MPM subjects, it is suggested as a potential circulating marker [107].

##### Toluene

It has been reported that toluene increases *CYP2E1* mRNA and modifies its protein activity on leukocytes. A venous blood study was performed on workers exposed to toluene in the air and twenty-four administrative workers in Guanajuato, Mexico. This study indicated that the methylation of *CYP2E1* and *IL6* promoters significantly increased in the group exposed to toluene. In addition, the levels of methylation in the promoter region of *CYP2E1* were higher in smokers exposed to toluene in comparison with non-smokers. A significant correlation was also found between the methylation of *CYP2E1* and the methylation at the promoter region of *GSTP1* and *SOD1* [108]. A recent study proposed the long noncoding *NEAT1* and a panel of four miRNAs (*miR-301a-3p*, *miR-16-5p*, *miR-15b-5p*, and *miR-15a-5p*) as an efficient epigenomic signature related with toluene-induced neurodegeneration [109]. Conversely, another study showed that the effects of chronic combined exposure to volatile organic compounds (VOCs, toluene, ethylbenzene, and xylene) are more unfavorable than those resulting from a single compound exposure [84]. Specifically, *MALAT1* and eight coding genes (*CACNG8*, *CLIP2*, *CNTNAP3*, *FMR1*, *GLS2*, *SULT4A1*, *TP73*, and *WNT7B*) expressions were significantly reduced by DNA hypermethylation in VOC-exposed subjects [110].

##### Radon

Recently, ncRNAs have been highlighted as early biomarkers of toxicity to radon exposure. In a cohort study performed with 144 miners, potential biomarkers were analyzed. Hematological parameters were statistically different in the underground group, while the levels of certain cyclins and cyclin-dependent kinases (e.g., CDK2, CDK4, CDK6) were higher. A miRNA microarray screening showed that five miRNAs (among them *miR-19a*, *miR-335*, and *miR-451a*) were downregulated (>2 fold) in the underground group, suggesting that specific miRNAs can be used as potential indicators of radon-mediated injury [111].

##### Mercury

Mercury (Hg) is a metal widely used in the industry [85]. Inhalation of mercury vapor can harm the nervous, renal, gastrointestinal, respiratory, and immune systems [112]. A recent study revealed an increased expression of histone deacetylase 4 (*HDAC4*), transcription factors specificity protein 1 (SP1) and 4 (SP4) in neuroblastoma (SH-SY5Y) Hg-treated cells. The results confirmed that the siRNA-mediated knockdown of MAPK14 (P38), SP1, SP4, HDAC4, or BDNF blocked Hg-induced cell death significantly. All these results suggested that MAPK14/SP1-SP4/HDAC4/BDNF may represent a new effector pathway involved in Hg-mediated neurotoxicity [113].

##### Cigarette Smoking

The vulnerability to cigarette smoke (CS) is one of the significant threats to human health worldwide. Numerous chemical substances present in tobacco contribute to its toxicity: PAHs, N-nitrosamines, heavy metals (e.g., cadmium, arsenic), aromatic mines, and alkaloids (e.g., nicotine and its metabolite cotinine) [114]. For decades, it has been known that smoking considerably increases the risk of cardiovascular diseases, different cancer types, as well as respiratory diseases [115]. CS is a complex chemical mixture containing thousands of compounds, several known carcinogens, co-carcinogens, and mutagens [12]. CS induces a significant increase in the expression of several chromatin modification enzymes, including DNA methyltransferases, histone acetyltransferases, histone methyltransferases, SET-domain proteins, kinases, and ubiquitinases [115]. Moreover, several epigenome-wide association studies (EWAS) identified a correlation between modifications in DNA methylation in smokers’ blood and their smoking status [116]. With regard to ncRNA regulation, several studies report that cigarette smoke extracts increase the HOX transcript antisense RNA (*HOTAIR*) levels in human bronchial epithelial (HBE) cells. *HOTAIR* mediates epithelial-to-mesenchymal transition (EMT), a process involved in the malignant transformation of cells [12,117,118]. In addition, *HOTAIR* induces epigenetic silencing of *CDKN1A* via enhancer of zeste homolog 2 (EZH2)-mediated tri-methylation of Lys 27 of histone H3, which contributes to cell cycle changes [116]. Similar studies indicate that the lncRNA *MALAT1* is also involved in cigarette smoke-induced EMT and malignant transformation in HBE cells [119].

### 3.2. Toxics in Water

The quality of water sources widely varies depending on local geology, agricultural activities, and industrial or municipal sources of pollution. As a rule, groundwater contains more mineral pollutants than surface water due to its passage along rocked beads on the earth’s surface, which are mineral rich. On the other hand, surface waters tend to contain more biological pollutants and more organic pollution than groundwater. Water sources can include both organic and inorganic industrial wastes [120,121]. Contaminants getting into drinking water can do it in a variety of ways. One is through pipeline corrosion which releases metals such as lead (described in Section 3.1.3), copper, cadmium, iron, zinc, nickel, and others into the water supply. In addition, there are problems caused by the by-products of chlorine when used as a disinfectant, being the most important the organic molecules called Trihalomethanes (THMs), of which the best known is chloroform. THMs are produced by naturally reacting chlorine with organic matter such as rotten leaves or soil and with industrial pollutants of organic origin [122,123].

#### 3.2.1. Arsenic

Arsenic (As) is a natural environmental toxicant found in drinking water, soil, and food [124]. Chronic exposure to As is involved with various tumorigenic events, including those from the respiratory tract and the bladder [125]. In human cells from the embryonic kidney (HEK239T) and cervix-carcinoma (HELA), As reduced global acetylation of lysine 16 of histone H4 (H4K16) in a dose-dependent manner [125]. As a possible mechanism, arsenic decreased histone acetyltransferase (HAT) by direct binding to cysteine into the Cys2HisCys (C2HC) zinc finger domain [125]. In addition, the analysis of CD4+ cells from women exposed to arsenic showed that individuals with high urinary arsenic concentrations had significantly lower levels of global trimethylated Lys-9 of histone H3 (H3K9me3) compared to women with lower urinary arsenic concentrations. The authors suggested that decreased histone methylation caused by As could lead to the expression of genes involved in tumorigenesis [126].

#### 3.2.2. Cadmium

Cadmium (Cd) is a human carcinogen [127] derived from industrial activities. The primary source of Cd exposure is food intake [128]. In human bronchial epithelial cells (16HBE), Cd inactivates tumor suppressors and genes associated with DNA repair through the hypermethylation of DNA, as a result of the elevated expression of DNA methyltransferases (*DNMT1* and *DNMT3A*) [129]. Furthermore, human lymphocyte cells treated with Cd increased their proliferation, expression levels of the DNA methyltransferases *DNMT1* and *DNMT3B*, and global methylation. In contrast, transcription levels of cyclin-dependent kinase inhibitor 2A (*CDKN2A*) were reduced compared to control cells. Still, its expression levels were rescued using a DNA methyltransferase inhibitor (5-aza-dC) with a concomitant reduction in proliferation even with Cd treatment. These data suggested that the aberrant increase in the Cd-exposed lymphocytes is induced by hypermethylation of the *CDKN2A* promoter [127].

#### 3.2.3. Chromium

Chromium (Cr) is a toxic and carcinogenic contaminant that directly reacts with DNA, forming adducts and inducing mutations. Cr compounds cause lung, gastrointestinal, and skin cancer [130]. New research studies show that human bronchial epithelial cells (BEAS2B) treated with Cr increased the levels of *NUPR1* mRNA and a reduction in acetylated Lys-16 of histone 4 (H4K16ac). Significantly, overexpression of *NUPR1* induces anchorage-independent cell growth, while its knockdown prevents Cr-induced cell transformation. These findings suggest a deleterious epigenetic mechanism of Cr through the induction of *NUPR1* expression and the decrease in H4K16 acetylation [130]. The promoter of *MLH1*, an essential DNA mismatch-repair protein, showed slightly increased methylation after Cr exposure in normal lung tissue. In tumor tissues exposed to Cr, the methylation of the *MLH1* promoter was remarkably higher [131]. These findings suggest that Cr-mediated carcinogenesis drives epigenetic changes in *MLH1* in normal airway epithelial cells. The methyltransferase SUV39H1 might be involved in the hypermethylation process in promoters followed by the accumulation of mutations in DSB repair genes such as *RAD50*, which could lead to lung cancer progression [131].

#### 3.2.4. Nickel

Nickel (Ni) is a toxic and carcinogenic metal used in industry [132]. The primary route of human Ni exposure is inhalation; thus, it causes multiple respiratory conditions such as irritation, inflammation, edema, and cancer [132]. A recent study showed that both epigenome and transcriptome undergo robust changes after the termination of the nickel exposure in BEAS-2B cells [133]. Upregulated transcripts correlated with a significant increase in the levels of H3K4me3 and the loss of repressive H3K27me3 after Ni exposure. These H3K4me3-gain and H3K4me3-loss regions correlate as well with several transcriptional activators and repressors, suggesting that these loci are important regions during dynamic chromatin changes [133].

#### 3.2.5. Copper

Copper (Cu) is a vital trace element within the human body, but it is also toxic [134]. The main sources of Cu from food include vegetables, potatoes, beef, and nuts. A group showed an association between DNA methylation at specific loci and Cu exposure by correlating Cu-related CpG sites, lower high-density lipoprotein cholesterol (HDL-C), and higher C reactive protein (CRP) levels, which are cardiovascular risk factors. This evidence suggested that the DNA methylation profile after Cu exposure modifies lipid metabolism and inflammation [134]. However, experimental evidence is necessary to support these observations and verify the specific mechanism by which Cu increases DNA methylation.

#### 3.2.6. Trihalomethanes

Trihalomethanes (THM) are undesired disinfection by-products (DBPs) formed during water treatment. Cytotoxicity of these compounds has been associated with chloroform and trichloroacetic acid exposure. Studies have reported changes in the epigenetic landscape due to global DNA methylation and proto-oncogenes hypomethylation after exposure to THM, as described in rodents. However, there is limited human evidence of DNA methylation changes in retrotransposons or other regions of the genome [135,136].

### 3.3. Toxicants in Food

Nutrition is one of the most studied and better understood environmental epigenetic factors [137]. It is proposed that both dietary nutrition and exposure to environmental toxicants are two factors that are intrinsically intertwined in health outcomes [138]. Dietary compounds are thought to effectively prevent epigenetically dysregulated conditions [139]. Nutrients can directly inhibit the activity of epigenetic enzymes such as DNMT, HDAC, or HAT or can alter the substrate availability necessary for their enzymatic activity. Consequently, it modifies the expression of critical genes and impacts overall health and longevity [137].

Food is a significant source of environmental toxicants, including bisphenol analogs, phthalates, PFASs, polychlorinated biphenyl (PCBs), particulate matter, parabens, heavy metals, as well as other emerging contaminants. Due to their hydrophobicity, many of these agents accumulate in fats during food processing and consequently in the fatty tissues of animal bodies [138]. Thus, variations in food consumption and the resultant exposure to additional levels of toxicants may vary, particularly in “Western diets”, which contain high protein and fat amounts [138]. In addition, food additives such as nitrates are related to an increased cancer risk from digestive organs. However, intake of nitrates from diet and other sources such as drinking water was not associated with these types of cancer [140].

In recent decades, it has been suggested that microbiota influence gene expression. However, we are still in the first stage of the discovery of biological processes in bacteria that regulate the expression of eukaryotic genes within the host in the context of metabolic diseases [141]. Furthermore, the relationship between the microbiota and the induced epigenetic programming has been recently explored. For example, volatile short-chain fatty acids have been proposed as mediators in the host [142]. However, most of the molecular processes that favor epigenetic changes remain to be deeply studied. Despite water quality being affected by increasing microbial pollution, how microbial agents would affect our epigenetic reprogramming is unclear.

#### 3.3.1. Glyphosate

N-(phosphonomethyl) glycine (glyphosate) is one of the ingredients in herbicides formulation, whereas the primary source of exposure is through its residues in food. Some studies have associated glyphosate-based herbicides with non-Hodgkin lymphoma (NHL), acute myeloid leukemia (AML), and endocrine-disrupting activities [143]. In this context, human peripheral blood mononuclear cells (PBMCs) incubated with glyphosate modified its global methylation profile, showing low expression of *CDKN2A* and *TP53*, and high expression of the cell cycle drivers *CDKN1A* and *BCL2* compared to untreated cells [143]. Similarly, PBMCs treated with glyphosate and its metabolite AMPA showed higher expression of genes involved in the DNA methylation process and deacetylation of histone proteins [144]. Therefore, glyphosate and AMPA exposure was related to expression changes in chromatin remodelers in these cells. In parallel, it is essential to evaluate the mechanism that induces these changes at the transcriptional level and its implications at the cellular level. On the other hand, in human embryonic kidney 293 (HEK293) cells treated with glyphosate, the transcription of cell cycle progression (*EGR1*, *AP1*, *FOS*, and *MRTL*) and regulation genes (*CCNB1*, *CCND1*, and *CKN1A*) was activated. According to this evidence, glyphosate enhances cell proliferation [143].

#### 3.3.2. Dioxin

Dioxin (2,3,7,8-tetrachlorodibenzo-p-dioxin) is a persistent and environmental toxicant that accumulates in the fat tissue due to its lipophilic nature [145]. In mice treated with dioxin, cardiomyocyte-like *Nkx2-5*^+^ cells were hypomethylated during cardiogenesis affecting cell signaling and cardiovascular development functions. On the other hand, embryonic stem cells did not show any changes in their methylation status in response to dioxin. The authors of this work suggested the impaired activity of TET proteins because of dioxin exposure [146]. The authors found 4821 genes differentially expressed in the same position, only 240 genes differentially methylated, and 111 genes that showed a significant correlation between DNA methylation and gene expression [146].

#### 3.3.3. Bisphenol A

Bisphenol A (BPA) is an organic synthetic compound widely used in plastic manufacturing, packaging of food and drinks, medical devices, thermal paper, and dental materials. BPA represents a contamination source in air and soil, as well as in beverages and food. It accumulates in several human tissues and is potentially harmful to human health [145,147]. BPA is also a xenoestrogen that acts as an endocrine disruptor by binding multiple targets inside and outside the nucleus. Furthermore, it is a weak carcinogen that may have long-term effects on the reproductive system by altering the epigenome [145]. In neonatal rats, BPA promoted phosphatidylinositol 3 kinase (PI3K)/AKT signaling activation by inducing histone methyltransferase EZH2 inactivation and the consequent reduction in the levels of trimethylated Lys-27 of histone 3 (H3K27me3) [148].

#### 3.3.4. Phthalate

Di(2-Ethylhexyl) phthalate (DEHP) is a plasticizer in polyvinyl chloride (PVC)-based materials. Exposure to DEHP often occurs via oral ingestion, inhalation, and dermal absorption, associated with reproductive alterations [149,150]. Upregulation of essential adipogenesis-related genes (*Pparγ*, *C/Ebpα*, and *Fabp4*) and acetylation of Lys-9 of histone 3 (H3K9ac) has been studied to be dose-dependent in mouse mesenchymal stem cells treated with benzyl butyl phthalate (BBP) [151]. In addition, these authors validated the downregulation of the histone methyltransferases *Setdb1* and *G9a* (responsible for the di-methylation of H3K9) and the upregulation of the histone acetyltransferases *Gcn5* and *Ep300*, along with the histone deacetylases *Hdac3* and *Hdac10.* Notably, *Pparγ* knockdown reverted these epigenetic modifications in mesenchymal cells. Hence, it seems that BBP enhances the adipogenesis of MSCs by altering the balance between histone modifications through *Pparγ* [151]. Therefore, it will be essential to determine a specific interaction mechanism between BBP and Pparγ that promotes the observed changes in epigenetic regulators.

### 3.4. Toxics in Consumer Products

Increasing amounts and varieties of chemicals are used in consumer products, while our understanding of their exposure pathways and associated human health risks still lags [152]. The following sections will summarize the epigenetic alterations related to some of the most used toxic agents.

#### 3.4.1. Nanotechnology

In the last decade, the nanotechnology field has become increasingly relevant in different aspects of human life [153]. Nanomaterials (NMs) are natural or engineered materials with one or more external dimensions in the size range of 1–100 nm [10,154]. Nanoparticles (NPs) and NMs are used in many daily life products including medicines, household items, food, toys, among others [153,155]. The increasing usage of NMs increases the probability of human exposure, favoring the considerable risk of nanotoxicity. “Toxico-omics” research has been performed on these materials. However, the epigenetic modulations in humans originating from exposure to these toxicants have not been widely studied [156]. Because of their ability to enter the human body via inhalation and skin penetration and subsequently interact with intracellular structures, several concerns over their potential toxicity to workers and end-users have been raised [10]. In addition, their effects on epigenetic markers remain a novel area in the field of nanotoxicology, with limited and inconclusive information, as well as several knowledge gaps [157,158]. The epigenetic aspects of the effect of nanomaterials on cells help to provide a better understanding of when and how nanomaterials may impact genome integrity and guide application strategies for nanomaterials in biomedicine. However, there is limited information about the effects of NPs (e.g., SiO_2_, AuNP, AgNP), quantum dots, and copper oxide (CuO) on the epigenetic landscape [159].

It is important to highlight that besides ncRNAs, histone post-translational modifications (PTMs) including methylation and acetylation are covalent modifications that have an essential role in gene regulatory function by modulating chromatin structure [8,45]. Generally, the repressive state of chromatin is enriched in H3K9 and H3K27 trimethylation (H3K9me3 and H3K27me3), while H3K4me3 is enriched at promoter regions and associated with more accessible chromatin [160]. Besides methylation and acetylation, other modifications such as phosphorylation of histones are indicative of toxicity-induced damage or environmental stress [161]. In the context of nanotechnology products, NMs and NPs are reported to induce disruption of the normal patterns of histone modifications and changes in DNA methylation. One of the most common alterations generated by prolonged exposure to a range of NMs and NPs is increased phosphorylation of histone H2AX at serine-139 (γH2AX) [155]. γH2AX is induced in response to various types of DNA damage and it is one of the earliest biomarkers of DNA damage in different pathologies [162,163]. In most of these reports, the induction of oxidative stress was carried out in parallel with the increase in the formation of γH2AX. Nevertheless, the oxidative stress independent of the phosphorylation of histone H2AX was also shown [155]. Recently, the effect of ncRNAs in association with different engineered NMs in an exposure model in mouse airways has been reported [164].

##### Titanium Dioxide

Nanoscale titanium dioxide (TiO_2_) is manufactured on a large scale, with various applications in consumer products. Different human cell lines (Caco-2, colorectal; HepG2, liver; NL20, lung; and A-431, skin) were exposed to TiO_2_ nanoparticles to assess the effects on global methylation. The reduction in global levels of DNA methylation and the expression of the DNA methyltransferases *MBD2* and *UHRF1* were demonstrated in A431, Caco-2, and HepG2 cell lines exposed to TiO_2_ nanoparticles. Eight genes (*BNIP3*, *CDKN1A*, *DNAJC15*, *GADD45A*, *SCARA3*, *GDF15*, *INSIG1*, and *TP53*) were proven to be methylated in promoter regions. The altered expression of these genes is directly associated with the etiology of different diseases. The results also revealed aberrant expression of epigenetic regulators implicated in DNA methylation (*MBD2*, *DNMT1*, *DNMT3a*, *DNMT3b*, and *UHRF1*) in cells exposed to TiO_2_ [165].

##### Gold Nanoparticles

Gold nanoparticles (AuNP) have great potential for various biomedical applications, but their extensive application is dependent on possible toxicity in the liver, which is their leading accumulation site. A study conducted to determine the cytotoxic effects induced by AuNP of different sizes (15 nm to 60 nm), shapes (nanospheres or nanostars), and coating (citrate or 11-mercaptoundecanoic acid MUA) in human HepaRG cells and rat hepatocytes, proved that AuNPs have low toxicity in liver cells. Among all AuNPs tested, the most miniature 15 nm spheres showed the highest toxicity [166]. Early observations indicated that the exposure of human fetal fibroblasts to AuNP affected the expression of 19 genes, for example, *miR-155* and *PROS1*, and induced global chromatin condensation [167]. Moreover, a most recent study which was performed in liver cancer cells demonstrated the altered expression of miRNAs (mainly miR-34a) in response to the application of AuNP [168]. Interestingly, both works concluded that AuNPs do not affect DNA methylation levels. Nevertheless, current studies aim to synthesize novel AuNPs for anti-cancer therapeutics focused on ncRNA inhibition [169].

##### Silica

Silicon-based materials are used in many applications, including drug delivery, dietary supplements, implants, and dental fillers. Silica nanoparticles (SiNPs) interact with immunocompetent cells and induce toxicity [170]. Recent studies proposed that these particles could alter gene expression by modulating epigenetic marks. The possible relationship between exposure to particles and these mechanisms could represent a critical factor in carcinogenicity. An investigation compared the effects of two transforming particles, NM-203 pyrogenic amorphous silica nanoparticle and Min-U-Sil^®^ (U.S. Silica Co., Frederick, MD, USA) 5 crystalline silica particle on the Bhas42 cell line. Short-term use of Min-U-Sil^®^ 5 decreased global DNA methylation and increased the expression of *DNMT3a* and *DNMT3b*, responsible for *de novo* DNA methylation. Treatment with NM-203 did not affect the expression of these enzymes nor DNA methylation levels. Furthermore, differences in global histone H4 acetylation status and HDAC protein levels were only observed in cells treated with Min-U-Sil^®^ 5. Finally, both types of particles induced a strong expression of *c-MYC* at the early stages of cell transformation, which correlated with the enrichment of RNA polymerase II and active histone marks in its promoter region [171].

##### The Balance between the Beneficial and Detrimental Effects of Nanomaterials and Nanoparticles

The main goal of nanotechnology is to develop smart nanomaterials that improve life quality without inducing adverse effects [172]. This fact explains why their wide usage and the global market are increasing in an exponential manner [173]. There are many pros and cons for their adoption; one of the most promising applications of NMs is in the field of medicine and pharmacology. For example, nanoparticles and nanomaterials have been designed as drug delivery systems and therapeutics. Besides their potential application as antitumor drugs [174], they can be used as carriers to deliver proteins, vaccine antigens, and drugs to a specific site of action [175]. A particular application in the context of ncRNA-based therapy (discussed in Section 5.10) is that nanoparticles have been developed for the safe transference of nuclei acids and increasing their stability to reach their targets [176,177]. Moreover, the application of NMs for the improvement of the environment is promising for the reduction in contaminants. NMs such as photocatalyst, nanosized adsorbents, nanomembranes, have been developed as emerging technologies for water purification [178].

On the other hand, NMs may have a double-sword effect. Their widely used application should be taken with caution as the exponential growth of nanotechnology usage imposes concerns over its impact on human and environmental health and safety (EHS) [173]. It is important to recognize that information about their environmental fate, transformation, transport, and accumulation in other environmental compartments (e.g., air, air, soil, and water) are still under study. Attention has been focused on their accumulation by aquatic environments [179]. Furthermore, there is evidence indicating that many NPs possess DNA-damaging activity; however, the conclusions are still controversial [173] As epigenetic mechanisms have an important role in the onset and progression of diseases, some attention should be given to obtain more inside correlation between epigenetic alterations and diseases, specifically for understanding the pathogenesis associated with nanomaterials. There are still some questions regarding the nano-size effect of materials (optical, thermal, magnetic, and the induction of reactive species, in the regulation of the epigenetic landscape, remain open. Taking both pros and cons of nanoparticles and nanomaterials, we consider that more research should be performed in the field of nanotoxicology to provide indications regarding their widespread usage in the future. As epigenetics represents a novel endpoint in the field of nanotoxicology [172], it should be important to increase the evidence for epigenetic toxicity in human-based biological models to distinguish between adverse health effects of NP exposure, in contrast to adaptative mechanisms to these nanoparticles. Possibly in the future, novel NMs could be developed to counterattack the effects of other NMs.

#### 3.4.2. Pharmaceutically Active Compounds

Pharmaceutically active compounds (PhACs) are a group of compounds that include hormones, antibiotics, and painkillers. They have adverse effects on numerous life forms because of their toxicity and can be found profoundly in wastewater (for instance, from hospitals) and surface waters [180]. Recently, the epigenetic effects caused by PhACs in aquatic animals have been investigated [181,182], but their consequences on human health remain unknown. In addition, very few studies have evaluated the effect of different epigenetic toxicant combinations. Multiple studies which assessed the impact of mixed industrial chemicals (e.g., p,a-DDE, Li, Fe, Ni, Sb, Pb, Bi, V, As, S) [183], mixtures of endocrine disruptors derived from plastics (BPA, DEHP, DBP) [184], and total xenoestrogens [185] on aquatic animals showed that PhACs influence DNA methylation. However, their impact on ncRNAs signature remains to be studied.

##### Diethylstilbestrol

Diethylstilbestrol (DES) is a synthetic nonsteroidal estrogen with high estrogenic activity [186]. DES induces a reduction in the expression of *P450scc* in the early stages of steroidogenesis. Recently, it was reported that the mouse cell line treated with DES, not only shows a reduction in the expression of *P450scc* but also induces histone modifications in its gene. This effect was further validated by ChIP assay, which confirmed the high occupancy of histone deacetylases on the *P450scc* promoter region upon treatment of TTE1 cell lines with DES. Such findings opened a novel panorama regarding epigenetic reprogramming caused by toxic compounds in a biological context associated with reproduction [186].

#### 3.4.3. Parabens

Parabens are preservatives widely used in consumer products, including cosmetics and food. In recent years, the effects of low doses of parabens on human health have been a matter of controversy [187]. The impact of prenatal paraben exposure on infant overweight was investigated by combining epidemiological data from a mother–child cohort with experimental approaches. Mothers who reported using cosmetic products containing parabens had elevated concentrations of them in their urine. Consistently, maternal exposure of mice to parabens increased food intake and weight in female pups. The effect was accompanied by an epigenetic modification in the neuronal enhancer of Pro-opiomelanocortin (*POMC1*), leading to reduced hypothalamic *POMC* expression [188].

#### 3.4.4. Aluminum Hydrochloride

Aluminum hydrochloride is a potential neurotoxic agent in cosmetic antiperspirants [189] which is associated with the risk of cognitive disorders such as neurodegenerative diseases. Moreover, aluminum exposure in rats was shown to cause spatial learning damage and impairs memory functions, mediated by increased expression of histone deacetylase 6 (*HDAC6*), reduced H3K9, and H4K12 acetylation at the promoter of the brain-derived neurotrophic factor (*BDNF)*, which will eventually inhibit *BDNF* expression [190].

## 4. Effects of Epigenetic Toxicants in Phenotypic Transformation

### Epimutations in Pollution-Associated Diseases

Every single toxicant described above displays harmful effects that differ among cell types. Consequently, various epigenetic changes trigger the activation, deactivation, or modulation of biological processes, which leads to the development of pollution-related diseases, including cancer. The following sections will describe the epigenetic implications focusing on ncRNAs as mediators of cellular responses upon exposure to toxicants in the human body. Furthermore, we will highlight seminal studies elucidating their relationship with pathological processes.

Glyphosate is extensively used in agriculture, forestry, urban areas, and aquaculture [191]. It is not approved for applications in the aquatic environment. Nevertheless, it can be easily detected in surface and groundwater [192]. Therefore, studies have indicated the presence of glyphosate in the human body, mainly in blood and urine [193,194], with cytotoxic effects in liver, lung, and nervous system [195]. In addition, this herbicide has been associated with alterations in methylation in the breast [196], blood [144], kidney [197], and lung fibroblasts [198]. In addition, it increases the expression of genes coding for enzymes that regulate DNA methylation processes, i.e., *DNMT1* (maintenance methyltransferase) and *DMNT3A* (de novo methyltransferase) in human PBMCs [144]. Increased DNMT1 and HDAC3 activities contribute to the condensation of chromatin, preventing transcriptional factors (TF) binding, resulting in the inhibition of transcription [144] related to neoplastic transformation. DNMT1 mediates the hypermethylation of *RASAL1*, which encodes an inhibitor of the RAS oncoprotein in primary human fibroblasts, causing their activation. Moreover, pathological hypermethylation by DNMT1, and therefore transcriptional dysregulation, can be induced by long-term exposure to the pro-fibrotic transforming growth factor TGF-β1, while increased *DNMT3a* expression induces methylome modifications in organ fibrosis [197]. However, it is currently unclear if glyphosate might promote the expression of *TGF-β1* in humans, although one report showed that glyphosate exposure induced inflammatory response in other species through *TGF-β1* upregulation [199].

Exposure to nickel (Ni), mainly insoluble Ni compounds, is related to the development of different types of cancer such as lung cancer [200] through disruption of epigenetic processes. Some studies demonstrated an alteration in methylation states in A549 cells due to an increase in di- and tri-methylation on H3K4 upon exposure to Ni [201]. Moreover, it is known that histone H4 has a high-affinity binding site for Ni in its N-terminal tail; thus, the exposure to Ni could lead to changes in the chromatin structure of target genes [202]. In addition, some ncRNAs may play a role in nickel-induced cell transformation and carcinogenesis; for example, *miR-21* is upregulated in mouse lungs by Ni nanocompounds exposure in a dose-dependent manner. *miR-21* enhances TGF-β/SMAD signaling events and finally promotes lung fibrosis [203]. Furthermore, in both in vitro and in vivo studies, mouse primary peripheral blood monocytes were exposed to Ni, which led to the upregulation of *miR-21* and further downregulation of *RECK. MMP-2* and *MMP-9* expression increased, and their activity resulted in excessive degradation of the ECM, causing reduced structural integrity [204]. Moreover, Ni and hypoxia inhibit the demethylase activity of JMJD1A and cause *SPRY2* repression in BEAS-2b cells [205]. Indeed, *SPRY2* downregulation contributes to cellular transformation in various human cancers, as in non-small cell lung cancer [206]. Chronic exposure to Ni also induces the upregulation of *ZEB1*, a gene associated with EMT that causes persistent downregulation of its repressors (e.g., *miR-205*, *miR-200a*, and *miR-200b*) [207]. Something similar occurs in rat lung tissues, where Ni oxide nanoparticles (NiO NPs) cause pulmonary fibrosis via activating TGF-β1 and a decrease in *MEG3* expression, a lncRNA that inhibits the levels of TGF-β1 and therefore the EMT occurrence [208].

Cadmium (Cd) can be ingested, inhaled, and absorbed by the skin because of its ubiquitous presence in the environment. Chronic exposure even in low doses has been related to kidney, liver, and cardiovascular damage, having adverse effects during reproduction and pregnancy [209]. Cd also has teratogenic solid and mutagenic effects due to aberrant alterations of multiple ncRNAs involved in pathophysiological conditions and signaling pathways that lead to the development of lung, breast, prostate, pancreas, urinary bladder, and kidney cancer, among others [210]. In vitro and in vivo studies have been performed to characterize some ncRNA signatures related to Cd exposure. For example, chronic Cd leads to a malignant phenotype in human prostate epithelial cells, where *miR-96* and *miR- 9* were upregulated; *miR-181d*, *miR-205*, *miR-125a-5p*, *miR-373*, *miR-146b-5p*, *miR-138*, *miR-155*, *miR-222*, *let-7b*, and *let-7c* were downregulated. These miRNAs have previously been related to pathways such as oncogenesis, cell survival, altered apoptosis, and cell adhesion [211].

Cd also affects miRNAs expression in the kidneys, specifically the renal cortex, leading to acute kidney injury, as seen in male Sprague Dawley rats treated with CdCl_2_, where expression of a nine-miRNA signature (*miR-21-5p*, *miR-34a-5p*, *miR-146b-5p*, *miR-149-3p*, *miR-224-5p*, *miR-451-5p*, *miR-1949*, *miR-3084a-3p*, and *miR-3084c-3p*) showed significantly increased levels. In contrast, the expression of *miR-193b-3p*, *miR-455-3p*, and *miR-342-3p* significantly decreased compared to control conditions [212]. Furthermore, patients with acute renal damage have higher *miR-21* levels in their urine than healthy individuals [212]. On the other side, Cd also transformed 16HBE cells. Among the miRNAs, *miR-27b-3p* appeared to be upregulated, while *miR-944* was downregulated in Cd-treated cells. Consequently, *CCM2* upregulation was induced. Additionally, it encodes a scaffold protein that promotes RHOA breakdown by interaction with SMAD. In endothelial cells, RHOA is necessary for the appropriate cytoskeletal organization, cell–cell interactions, and lumen formation [213].

In addition, Cd is highly present in cigarettes and upregulates the expression of *miR-101* and *miR-144* that target the *CFTR* 3′UTR to suppress its expression in 16HBE cells. This was also demonstrated in C57BL/6 female mice exposed to cigarettes, where an upregulation of *miR-101* resulted in the suppression of *CFTR* in the lungs [176]. Moreover, *miR-101* is more expressed in lung samples from patients with severe COPD than control patients [214]. In the same Cd-transformed 16HBE cells, a differential lncRNA expression has been reported, a 369-lncRNA signature was upregulated, whereas a ninety-lncRNA signature was downregulated. The lncRNA *ENST00000414355* was highly expressed, and its silencing reduced the expression of DNA damage-related genes (*ATM*, *ATR*, and *ATRIP*), while DNA repair-related genes increased [215]. In Sprague Dawley rats exposed to Cd, the expression of *ENST00000414355* corresponded with the expression of its target genes [215]. The same animal model showed a positive correlation between *MALAT1* and *BAX*, *TGF-β1*, and *STAT3* expression. Thus, *MALAT1* knockdown in Cd-transformed 16HBE cells inhibited cell proliferation, apoptosis, migration, and invasion [216]. Since the liver is the principal site for Cd deposition, several investigations have focused on liver harm by Cd exposure. A report demonstrated that at a high concentration of Cd in HepG2 cells, many miRNAs belonging to the *let-7* family (tumor suppressor miRNAs) were dysregulated [217]. Furthermore, in HepG2 cells, the lncRNA *MT1DP* is induced by Cd exposure, thus promoting cell death by activating RHOC-CCN1/2-AKT signaling and the consequent acceleration of Ca^2+^ influx, resulting in accelerated and elevated Cd toxicity [130]. *MT1DP* further enhances the repressive function of *miR**-365* that decreases NRF2 levels, a protein that responds to Cd-induced oxidative stress and participates in removing intracellular ROS in the HepG2 cell line, thus aggravating oxidative stress in hepatocytes [218].

Exposure to chromium (Cr) occurs mainly via inhalation or dermal absorption, although nowadays hexavalent chromium [Cr(VI)] is highly present as a toxicant in drinking water [219]. It has become a significant public health concern with toxic effects in the skin, oral, digestive, hepatic, respiratory, and renal systems [219], capable of inducing carcinogenesis through both genetic and epigenetic mechanisms, as evaluated in mice [220] and human cells [221]. People highly exposed to Cr may have high concentrations in the urine associated with the expression of *miR-21* and *miR-155*, promoting chronic kidney diseases [222]. In addition to urine, Cr has been detected in blood where it induces *miR-142-3p*, *miR-19a-3p*, and *miR-19b-3p* downregulation, and in contrast with upregulation of *miR-941*. Among them, *miR-19a-3p* is highly expressed in ovarian and gastric cancer, although the mechanism inducing Cr-mediated cellular transformation is unknown [223]. Decreased *miR-3940-5p* [224] and *miR-143* levels in plasma are also linked to high Cr concentration in blood. In BEAS cells exposed to Cr, *miR-143* was reduced, whereas *IL6* (a cytokine that regulates cell differentiation and proliferation) levels were significantly higher than control cells. Therefore, the overexpression of *miR-143* decreased the expression of *IL6*, inhibiting at the same time *HIF-1α* and *NF-κB P65* expression, thus affecting tumor growth [225].

In 16HBE-Cr cells, the relative expression of lncRNAs increased or decreased gradually in a dose-dependent manner. For instance, *RP11-817O13.6* increased first (0.63 μmol/L) and then decreased at the higher dose (10 μmol/L), suggesting that the ncRNA profiling may change according to the amount, the time of exposure, and even the cell type, as seen in BEAS-2B-Cr (1 µM) cells where *miR-143* expression levels were much higher than in the lung cancer cells A549 and H2195, that further lead to Cr (VI)-induced cell malignant transformation via upregulation of insulin-like growth factor-1 receptor (*IGF-IR*) and insulin receptor substrate-1 (*IRS1*) expression [226]. Moreover, the interleukin-8 is strongly induced by Cr (VI) through the activation of the IGF-IR/IRS1 axis, followed by the activation of the downstream ERK/hypoxia-induced factor HIF-1α/NF-κB signaling pathway, resulting in enhanced angiogenesis [227]. Other authors have suggested different Cr(VI) mechanisms such as the *miR-21-PDCD4* signaling axis [226] and by inducing cancer stem cell (CSC)-like properties with increasing *c-MYC* (target of *miR-494*) levels through the downregulation of *miR-494* at a chronic low dose (0.25 µM) [228].

All these mentioned works attempted to describe some action mechanisms of epigenetic toxicants mainly by in vitro experiments. However, in most of them, only a single ncRNA molecule is highlighted and represented as a unique mediator of crucial biological pathways that lead to the phenotypic transformation of the cell. This approach sets aside other essential factors such as associated proteins, methylation changes, participation of multi-component structures, and loci at specific times of the ncRNAs of interest. Therefore, the importance of conducting robust, unbiased, and integrative studies emerges to assess potential multi-omics technologies and the implementation of biochemical and molecular experiments able to describe epigenetic processes with high precision.

## 5. ncRNA Dysregulation in Pollution-Related Cancer

An increasing number of studies suggest that mutations or abnormal expression of ncRNAs are closely related to various diseases, including cancer [229]. Particularly, miRNAs have an oncogenic or tumor-suppressive function; even miRNA expression is globally suppressed in tumor cells compared with normal tissue, suggesting that miRNA biogenesis might be impaired in cancer [230] (Figure 3). For example, mercury chloride treatment in the human embryonic kidney (HEK-293) and human umbilical vein endothelial (HUVEC) cells induced the overexpression of *miR-92a* and *miR-486*. This work suggested that *miR-92a* and *miR-486* moderate the NF-κB pathway activation during mercury toxicity [114]. However, it is not yet clear how *miR*-*92a* and *miR*-*486* are upregulated in workers and cells after mercury exposure.

### 5.1. ncRNA Dysregulation in Liver Cancer

In HepG2 cells treated with Cd, the lncRNA *MT1DP*/RhoC interaction activates the CCN1/2-AKT pathway. MT1H competes with *miR-214* to avoid *MT1DP* suppression, triggering cell death. *MT1DP* decreases NRF2 activity, an antioxidant gene transcription factor, to evoke oxidative stress through the elevation of *mi**R-365*, which directly represses NRF2 transcription [208]. These studies reveal a new mechanism underlying the *MT1DP*-conducted signaling promoting Cd toxicity in hepatocytes.

The compound hexahydro-1,3,5-trinitro-1,3,5-triazine (also known as cyclomethylenetrinitramine, RDX) is an environmental toxicant resulting from human activities, such as military actions. Additionally, the effect on the global expression profile of miRNAs in B6C3F1 mice exposed to RDX was reported. Interestingly, the alteration in miRNAs profiles showed that the brain is the most affected organ upon RDX exposure, compared to the liver [231]. Moreover, RDX exposure increases the expression of *miR-206,* reducing *BDNF* and tankyrase 2, poly(ADP-ribose) polymerase tankyrase 2 (*TNKS2*) cluster gene expression potential targets in the brain, which may influence the neurotoxic effect by RDX. Moreover, many cancer-related miRNAs were significantly affected by RDX exposure [231].

### 5.2. ncRNA Dysregulation in Ovarian Cancer

A ncRNA signature may predict the origin of different histologic types of cancer even with a different tumor grade. In the case of ovarian cancer (OC), many authors have demonstrated specific miRNA and lncRNA expression patterns from various approaches. High-grade ovarian serous carcinoma (HGOSC) is the most common ovarian epithelial malignancy, and it shares similar histologic features with serous uterine carcinoma (USC) that make the diagnosis difficult [232]. A panel of 11-miRNAs, identified with a random forest model, could discriminate between OSC and USC, among them, *miR-196b*, *miR-532*, *miR-886-3b*, and *miR-99b* exhibited lower levels in HGOSC compared with USC, whereas *miR-29c*, *miR-155*, *miR-454*, *miR-146b5p*, *miR-486-5p*, *miR-27a*, *miR-192*, and *miR-141* exhibited higher expression in HGOSC. Moreover, a nested cross-validated 6-miRNA predictive model was able to classify the tumors with 91.5% accuracy [232]. Although other studies show different signatures with significantly up-modulated *miR-200* family and *miR-141* in a well-differentiated OC subtype, suggesting their role in developing a specific epithelial OC subtype [233]. Likewise, lncRNA signatures have been related to critical clinical aspects in patients with OC, for example, *LINC00861*, *LEMD1-AS1*, and *LOC101927151* may act as prognostic factors [234], and the expression of *lncSERTAD2-3*, *lncSOX4-1*, *lncHRCT1-1*, and *lncMYC-2* (also *PVT1*) are independent prognostic markers of relapse and survival for stage I OC [235]. Additionally, patients with high or low risk have distinctive lncRNA signatures [236]. In addition, the predictive value may be independent of other clinic-pathological factors, as in the case of the eleven-lncRNA signature validated for the prediction of both early and late survival [237]. Interestingly, other studies have established lncRNAs- and multi-RNAs-based models that could predict the response to treatment [238] and the recurrence risks [239], respectively.

### 5.3. ncRNA Dysregulation in Breast Cancer

Concerning breast cancer (BC), several investigations aimed to establish a ncRNA signature associated with specific characteristics of the different BC subtypes. For example, triple-negative breast cancer (TNBC), compared to other breast cancer subtypes, has the highest incidence and poorest prognosis. A logistic regression model identified an eight-miRNA signature associated with tumor recurrence and a decreased survival in high-risk patients [240]. Detection of BC at an early stage can prevent the development of metastasis and death. Therefore, another approach consists of characterizing molecular signatures based on tissue profiles and validating them with serum profiles [241]. In this way, four miRNAs have been validated for patients at the early stage, although in this case, most of the used samples were from Hispanic patients, while serum samples were from Chinese patients [241]. Putatively the lack of a uniform cohort may reduce the classification accuracy of the signature.

Using a support vector machine (SVM)-based classifier model, a thirty-four-miRNA signature could categorize patients into early and advanced BC stages with more than 80% accuracy, including four miRNAs associated with poor prognosis [242]. Clinical risk factors also contribute to prognosis prediction. A six-miRNA signature and risk factors as age, tumor-node-metastasis (TNM) stage, and some molecular characteristics (receptor for estrogens and progesterone, as well as human epithelial growth factor receptor 2-HER2 status) can predict 5-year overall survival (OS), with a predictive accuracy (0.75) higher than TNM stage (0.65) which is generally used for prognostic assessment and determine specific treatments [243]. Meanwhile, with the least absolute shrinkage and selection operator (LASSO) regression method, 17-miRNA-based OS, we set to predict a 13-miRNA-based recurrence-free survival (RFS) classifier signature [244].

With the same method, and highlighting the importance of the tumor immunology, eight immune-related lncRNAs correlated with OS [245] and a six-lncRNA signature could separate two immune risk subgroups, where cohorts of high risk showed reduced levels of CD8-T cells and the immunity-related signature compared to cohorts of low risk, suggesting a role of CD8-T cells as a prognostic feature [246]. In addition, another ten-lncRNA signature for RFS [247] and an eight-lncRNA signature that can divide a cohort into two risk groups with different survival times [248] have been further validated as independent predictors. A lncRNA signature may help predict the sensitivity and outcome response to chemical treatment, a tamoxifen efficacy-related lncRNA signature (TLS) was obtained to predict the response to tamoxifen of ER^+^ BC patients, demonstrating that patients with increased levels of TLS were more resistant to tamoxifen therapy as these lncRNAs interfered with cell cycle arrest induced by the drug [249]. Even a circRNAs signature is associated with docetaxel resistance in BC-cell lines where *circABCB1* was upregulated, whereas *circEPHA3.1* and *circEPHA3.2* are downregulated [250].

### 5.4. ncRNA Dysregulation in Pancreatic Cancer

Pancreatic cancer (PC) has the worst prognosis of all cancers. More than 90% of PC patients are diagnosed at late stages, suggesting that RNA signatures should have a potential role as biomarkers. Recently, a thirteen-miRNA signature was developed which helps to better evaluate and predict patients with a high OS [251]. Some miRNAs (*miR-103-2*, *miR-126*, *miR-340*, *miR-374b*, and *miR-627*) were associated in addition with other characteristics, such as alcohol consumption [251], suggesting the importance of integrating them into other clinical characteristics of the patients to obtain more specific signatures. The pancreatic ductal adenocarcinoma (PAAD) is the most frequent subtype where a five-miRNAs panel (*miR-125a*, *miR-328*, *miR-376b*, and *miR-376c*, *miR-1301*) could increase the capacity of prognostic prediction for patient survival [252]. Meanwhile, by a prospective study for longer follow-up times (which may represent tumors in earlier stages of tumorigenesis), positive associations between the over-expression of four-miRNAs (*miR-10a*, *miR-10b*, *miR-21-5p*, and *miR-30c*) and PAAD development may be set [253]. Although other two-miRNAs (*miR-33a-3p* and *miR-320a*), in addition to the serum carbohydrate antigen 19.9 (CA19.9), used for monitoring response to treatment, increased the identification of patients with PC or premalignant intraductal papillary mucinous neoplasm (IPMNs) from healthy people with an accuracy of 0.95 [254]. In another study, the application of a three-miRNA signature (*miR-125a-3p*, *miR-5100*, and *miR-642b-3p*) for PC diagnosis yielded the same accuracy [255]. However, through a LASSO regression model, a five-lncRNA signature was identified which was shown to be correlated with better survival [256]. In another study, a four-lncRNA signature (*ABHD11-AS1*, *HOPPIT*, *LINC00460*, and *miR-205HG*) demonstrated to have a negative correlation with OS, whereas a two-miRNA signature (*C9orf139* and *miR-600HG*) correlated to OS in a positive manner [257].

### 5.5. ncRNA Dysregulation in Colorectal Cancer

Several lines of evidence indicated that genes associated with the immune response together with lncRNAs serve as predictor markers for OS in colon cancer patients. Additionally, Lin et al. identified a lncRNA signature associated with the immune response for prognosis in patients. A highly expressed lncRNA signature (*AC008760.1*, *AC009237.14*, *AC083809.1*, *AL391422.4*, *AL445645.1*, *LINC01063*, and *LINC01234)* was associated with poor prognosis; on the contrary, the expression of *AC016027.1* indicated an association with good prognosis [258]. Pichler et al. analyzed genome-wide miRNA sequencing data of 228 colorectal cancer patients from The Cancer Genome Atlas dataset to identify miRNAs significantly associated with survival. A six-miRNA signature (*miR-92b-3p*, *miR-188-3p*, *miR-221-5p*, *miR-331-3p*, *miR-425-3p*, and *miR-497-5p*) was identified strongly predicting patient survival in the screening cohort. High *miR-188-3p* expression proved to be an independent prognostic factor since its exogenous expression increased migration of cancer cells in vitro and the formation of metastases in vivo. Mechanistically, the pro-migratory role of *miR-188-3p* is mediated by direct interaction with MLLT4, a novel biomarker associated with cell migration in colorectal cancer [259].

### 5.6. ncRNA Dysregulation in Lung Cancer

*MALAT1* is one of the most assessed lncRNAs involved in important biological processes. In fact, numerous molecules, and their corresponding signaling pathways, have been validated as targets of *MALAT1* in lung cancer. *MALAT1* activates the epithelial-mesenchymal (EMT) process and increases brain metastasis-initiating lung cancer cells [260]. In addition, *MALAT1/miR-101-3p*/MCL1 axis mediates cisplatin sensitivity in cisplatin-resistant A549/DDP cells [261]. Another vastly assessed oncogenic lncRNA in lung cancer is *XIST* (X-inactive specific transcript), which was reported to be increased in NSCLS cell lines compared to normal human bronchial epithelial cell lines. Concordantly, *XIST* silencing suppressed proliferation, invasion, migration, and finally programmed cell death [262]. Furthermore, the association between expression levels of lncRNAs panels and patient prognosis has been evaluated in several investigations. For example, a high-throughput study has shown the potential of *ADAMTS9-AS2*, *AC011483.1*, *TTTY16*, *AC006238.1*, *LINC00462*, and *CACNA2D3-AS1* in the prediction of OS of patients with lung cancer [263]. Gong et al. have recently compared RNA-seq data with the TCGA repository to find lncRNAs with predictive potential in lung cancer. This analysis identified 84 dysregulated lncRNAs in lung cancer tissues, among which ten lncRNAs were mainly associated with the survival of patients. In addition, *LINC01537* has been recognized as the most significant lncRNA whose downregulation was verified in the subsequent analysis [264].

Up to now, numerous investigations exist analyzing lncRNA expression data in lung cancer, and robust validations in non-dependent datasets. However, the lncRNA landscape in NSCLC is far from comprehensive, mainly because of the minor independent validation inconsistency in targets and sample sets [259]. Recently, Acha-Sagredo et al. validated a twelve-lncRNA signature (upregulated *FEZF1-AS1*, *LINC00673*, *LINC01214*, *LINC01929*, *NUTM2A-AS1*, and *PCAT6*; and downregulated *ADAMTS9-AS2*, *FENDRR*, *LANCL1-AS1*, *LINC00968*, *PCAT19*, and *SVIL-AS1)* which is aberrantly expressed in NSCLC cancer. In addition, abnormal DNA methylation was observed in the promoters of genes such as *FENDRR, FEZF1-AS1,* and *SVIL-AS1. FEZF1-AS1* and *LINC01929* improved patient survival, according to the Cancer Genome Atlas (TCGA) set [265]. In a different approach focused on miRNAs, human bronchial epithelial cell line (BEAS-2B), as well as lung cancer tissues from patients exposed to arsenic showed an apparent, sustained upregulation of *miR-301a* by IL6/STAT3/SMAD4 signaling pathway, which contributed to carcinogenesis after chronic exposure of arsenic [126].

### 5.7. ncRNA Dysregulation in Kidney Cancer

Renal cell carcinoma (RCC) is the most common malignant tumor of adult kidneys (3.7% of all adult cancers worldwide) [266]. Although lncRNAs are seen as promising biomarkers for cancer initiation and progression, their prognostic significance in RCC is not fully understood. Liu et al. determined the ability of lncRNAs as predictors of poor prognosis in clear cell RCC (ccRCC). A total of 525 patients were analyzed based on the cohort of kidney renal clear cell carcinoma (KIRC) on the TCGA repository. Hierarchical clustering of the KIRC cohort allowed the identification of 26 differentially expressed lncRNAs (11 downregulated and 15 upregulated lncRNAs). Further survival analysis identified a significant thirty-lncRNA signature for the prediction of ccRCC prognosis, with four differentially expressed lncRNAs (*HAR1B*, *miR155HG*, *PVT1*, *and TCL6*) that correlated with OS [266].

In addition, the function of miRNAs in oncogenesis has been studied in RCC cells. Notably, it was revealed that tumor suppressor genes such as *APC*, *MEG3*, and *PTEN* are targets of oncogenic miRNAs such as *miR-7*, *miR-301a*, *miR-22*, *miR-193a-3p*, and *miR-671-5p*, suggesting a role of miRNAs in the pathogenesis of RCC. In contrast, tumor suppressor miRNAs (downregulated in RCC cells) have functional roles in the induction of apoptosis and cell cycle arrest. Genes associated with epithelial-to-mesenchymal transition (EMT) such as *HIF-1a*, *HOTAIR*, *SLUG*, and *ZEB1* are targets of this tumor suppressor miRNAs. Therefore, their downregulation, in consequence, promotes the enhancement of EMT [267]. Furthermore, signatures based on miRNAs can be used for the classification of subtypes in RCC. For example, Youssef et al. classified different subtypes of RCC (e.g., clear cell, papillary, oncocytoma, and chromophobe RCC) with sensitivity values [268].

### 5.8. ncRNA Dysregulation in Acute Leukemia

Acute lymphoblastic leukemia (ALL) is known as the most common form of pediatric leukemia. One of its main characteristics is the rapid growth of abnormally developing lymphoid cells within the bone marrow and the reduced production of normal blood cells [269]. Based on RNA sequencing approaches, different studies demonstrated that lncRNAs and miRNAs are crucial in the pathogenesis of progenitor B-cell acute lymphoblastic leukemia (B-ALL). Even though the number of studies is limited, they indicate that the expression of lncRNAs is modified in this type of leukemia [231]. For example, Dinesh Rao and colleagues found “B-ALL-associated long RNAs” or BALR, differentially expressed in B-ALL (*BALR-1*, *BALR-6*, *BALR-2*) as well as *LINC00958*. Particularly, the lncRNA *BALR-2* was correlated with poor OS and poor response to the pharmacological treatment (prednisone) [269,270].

In addition, miRNAs have vital biological importance during the development of B-cells and may contribute to associated malignancies. For example, the miRNA database from ALL and LeukmiR predicted 861 miRNAs associated with the disease [271]. Furthermore, the expression of miRNAs can discriminate between the different types and subtypes of leukemia. For example, a twenty-seven-miRNA signature was identified in 72 patients, which was differentially expressed among ALL and acute myeloid leukemia (AML) samples [272]. In another study, a four-miRNA signature (*let-7b*, *miR-128a*, *miR-128b*, and *miR-223*) discriminated against two leukemic types. A subsequent study also confirmed that *miR-128a* and *miR128b* (in addition to *miR-34a*, *miR-213*, *miR-210*, *miR-130b,* and *miR-146a*) were differentially expressed and played a significant role in ALL [273]. In addition, miRNA signatures were useful to predict prognosis and relapse. Upon miRNA profiling using paired samples, Han et al. identified a three-miRNA signature (*miR-27a*, *miR-223*, and *miR-708)* in children with ALL-associated relapse [274]. Similarly, other miRNAs (*miR151-5p*, *miR-451*, and *miR-1290*) have been identified as a putative signature for relapse occurrence when the miRNAs profile correlated with patient diagnosis and clinical outcomes [275]. Beyond miRNAs and lncRNAs, other ncRNAs have implications in leukemia, such as circRNAs [276] and snoRNAs, although they have not been extensively studied in B-ALL. Nevertheless, they have been proposed as biomarkers in hematological diseases [269].

### 5.9. ncRNA Dysregulation in Testis Cancer

Testis cancer is the most prevalent solid tumor in young men (15 to 40 years old), with a constant increase in the frequency of the cases [277]. LncRNAs that alter testicular tumor pathogenesis such as *XIST*, *H19*, *SPRY4-IT*, *NLC1-1C*, and *HOTTIP*, but most of them have not been evaluated in clinical studies, except for *XIST*. Previous reports have suggested the expression of the lncRNA *XIST*, and epigenetic marks related to the inactivation of the X chromosome as specific biomarkers [277]. In particular, the demethylation of *XIST 5′* is augmented in seminomas compared to non-seminomas or normal tissue. In consequence, this biomarker has a potential value for the assessment of the quality of spermatogenesis in individuals [278].

Besides lncRNAs, the *miR-371-373* cluster has been suggested as putative markers in patients with testis cancer. Some groups evidenced that *miR-302/367* and *miR-371a-3p* and expression significantly differs between testicular tumors and healthy testicular tissue. Several clinical studies verified that these miRNAs greatly improved the sensitivity and specificity of conventional markers [279]. However, all these studies have certain limitations that need to be addressed. Although advanced bioinformatic methods were implemented, most reviewed papers did not describe the quality of the RNA-seq dataset used; in some works, available microarray was downloaded. Additionally, other researchers assessed miRNAs expression by a single- or multiplex qRT- PCR methodology. Moreover, only a few explained the population characteristics from which the sequencing data were obtained, mainly ethnicity. On the other hand, these works mention that data were obtained mainly from Asian or European populations. Then, this implies that the proposed signature may not be strictly valid for a global population, only for specific ethnic groups or none. Furthermore, most authors mention that their validation method should be improved by increasing the sample size, using data from prospective studies, and integrating into vitro or in vivo experimental data to confirm the results. The experimental validation is crucial to determine a more precise signature. As expected, this needs to be unbiased and integrative in the sense of making use of other kinds of omics-derived, high-quality sequencing data. In addition, the simultaneous consideration of multiple clinical factors and already known biomarker tests proved in biofluids (plasma, urine, saliva, etc.) would enhance the precision of the potential signatures to achieve a high enough specificity [280] and finally be considered clinically useful.

### 5.10. Potential Role of ncRNAs as Therapeutic Agents in Exposome-Associated Diseases

Since ncRNAs have a functional role in modulating the expression of multiple downstream gene targets and associated pathways in cancer and other diseases, there is a rationale that suggests that ncRNAs-based therapies are promising as therapeutic options [176]. Different ncRNA-based strategies have been proposed for therapeutical purposes: small interfering RNAs (siRNAs), antisense oligonucleotides (ASOs), short hairpin RNAs (shRNAs), anti-microRNAs (anti-miRNAs), miRNA mimics, miRNA sponges, therapeutic circular RNAs (cirRNAs), and CRISPR-Cas9-based gene editing [281]. Several clinical trials have confirmed that ncRNAs are successful for the treatment of complex diseases including cancer [176,282]. According to a recent review by Winkler et al., 11 therapeutics based on ncRNAs have been approved by the FDA and/or the European Medicines Agency (EMA), and many other ncRNA approaches are under phase II or III clinical testing. The majority of the approved therapeutics correspond to siRNAs or ASOs that promote downregulation of genes, or that target pre-mRNA splicing. It is important to remark that there are no ncRNA-based strategies that have entered clinical testing so far [215]. However, recent investigations highlight their potential as ncRNA-based therapeutics. For example, Battistelli et al. studied a mutant form of the ncRNA *HOTAIR* to reduce the epithelial-to-mesenchymal transition (EMT). Their results showed that the *HOTAIR* mutant form was able to reduce the binding of *HOTAIR* to *Snail*, impairing EMT [283].

Although ncRNAs represent a novel therapeutical option, there are still some limitations to their adoption in clinical practice [184]. Trials have reported ambivalent results with some studies reporting positive effects whereas others showed limited success or inducing toxicity [185]. Issues related to specificity (undesired on-target effects), delivery, and tolerability hamper their translation into the clinical scenario [184,185]. However, novel approaches to overcome these limitations are under investigation such as chemical modification of the backbone of the nucleic acid sequence [284], and the use of nanoparticles for improving their delivery and stability as they offer protection against degradation by nucleases [146,285].

Several technological advances have increased the capacity to design and manufacture most ncRNA mimics. However, a clear limitation for their use is the current lack of deep understanding of their functional roles in oncogenesis and their downstream genetic targets [286]. For example, the therapeutic application of transfer RNAs, circular RNAs, small nucleolar RNAs, and piwi-interacting RNAs is still at its infancy [286].

## 6. Transgenerational Inheritance after Epi-Toxicants Exposure

Recently, the concept that parental experiences can shape the offspring’s behavior and physiology has stimulated new perspectives in the epigenetics field, with an emphasis on disease risk and resilience. In addition, epigenetic transgenerational inheritance has revealed that parents’ exposure to deleterious factors such as stress, toxicants, or abnormal dietary intakes can lead to germline-mediated inheritance of epigenetic modifications even without a constant impact on the environment [287].

The fundamentals of cellular reprogramming rely on epigenetic mechanisms beyond and above intrinsic DNA sequence; meaning that without epigenetic reprogramming, those epigenetic marks which are acquired during development or those externally induced by the environment, cannot be erased. If epigenetic reprogramming fails, such epigenetic marks can be transmitted from one generation to the next one. In other words, parental (F0 generation) exposure to toxic environments can influence germlines (female ova and male sperm) as well. This implies that the next generation (F1 generation) is still considered exposed, and the F2 generation will be the first transgenerational “unexposed” generation [287]. In mammals, these mechanisms took longer to be proved convincingly, compared to simpler organisms such as nematodes (*C. elegans*), where evolution seems to ensure across generations the deletion of potential harmful experiences during parental lifetime.

Currently used chemotherapeutic treatments have been proven to cause effects in the F1 generation, but not in the F2 generation since the main off-target effect of these drugs is exacerbated DNA damage [288]. A transgenerational phenotype, also known as a genetic trait, requires permanent germline reprogramming, mainly due to the methylation state. For instance, primordial germ cells (PGCs) acquire genome-wide de novo methylation within one day of development and subsequent migration into the genital ridge. However, after their entry, there is a rapid removal of DNA methylation of regions within imprinted and non-imprinted loci in males and female embryos [289]. Afterwards, germ cells in the gonad undergo a process in which they are methylated during gonadal sex determination [290].

The fact that an environmental factor (for example, an endocrine disruptor such as BPA) can reprogram the germline and promote a transgenerational disease state has significant implications for evolutionary biology and disease etiology. Although epigenetic inheritance is commonly associated with plant inbreeding [291], some pioneering studies describing this phenomenon in animals date back to the 1950s. Indeed, by the seminal studies of Conrad Waddington, who discovered in *Drosophila melanogaster* that up to seven generations display the exact wing structure change induced by heat shock in F0 [292]. Slowly, the definition suggested by Waddington shifted toward the concept of heritability. However, some reports of rodents appeared almost 50 years later. In 2005, Anway et al. demonstrated that the exposure of gestating rats to vinclozolin (an antiandrogenic fungicide) or methoxychlor (an estrogenic pesticide) induced an adult phenotype from F1 until F4 generations, including decreased spermatogenic capacity and increased incidence of male infertility. Interestingly, this work correlated DNA methylation patterns to the altered reproductive phenotype [293]. Other reports demonstrated that transgenes defined mice’s specific active or inactive transgenerational inheritance. For instance, the obese, diabetic, tumor susceptible, and yellow fur phenotype in mice is not only linked to the ectopic expression of the agouti protein. Still, it is directly induced by a transposable element located 100 kb upstream of the *agouti A* gene [294]. Even if this process is not likely conserved in humans, several epialleles for cancer predisposition (for instance, the *MLH1* locus for colorectal cancer or the *DAPK* loci for lymphocytic leukemia) depend on specific polymorphisms on the DNA sequence and are maintained in every generation. Moreover, these individuals display young-onset cancer by non-Mendelian inheritance through their reversal in the germline [295,296].

At this point, it is essential to set a clear difference between intergenerational, such as in utero exposure to nutrients, hormones, toxins at embryonic stages, and actual transgenerational mechanisms [297,298]. In addition, in subsequent generations, there may be an increased susceptibility to disease since they would respond in an aberrant manner to the same environmental agents, which therefore develops into generational toxicology, processes that usually involve transposable elements and are generally termed as “germline epimutations” [299].

Epigenetic marks in germ cells, which are vulnerable to environmental stimuli and capable of directing cellular fates during development, may mediate the effects of parental environmental exposures on the offspring’s behavior and physiology. Events of post-translational histone modification, DNA demethylation, and ncRNA expression panels in sperm have been implicated as transmitters of parental experiences (stress, nutrition, drug abuse). Rodent models examining paternal transmission have identified epigenetic signatures in mature sperm as possible substrates of transgenerational programming, namely patterns of (1) histone modifications (mostly at active marks such as H3K4me3, or inactive marks such as H3K9me2/3 and H3K27me3), (2) DNA methylation on CG, CH and CHH motifs, and (3) populations of noncoding RNAs (miRNAs, siRNAs, piRNAs, tRNAs, and lncRNAs). Therefore, different RNA subtypes are of primary interest, as they may be altered through intercellular communication via epididymosomes even in transcriptionally inert mature sperm, where DNA condensation impedes other epigenetic changes [300]. However, the identity of specific ncRNA signatures in sperm responsive to environmental clues is not yet clear.

Consistent evidence in rodents has suggested the impact of transgenerational inheritance in complex diseases as hepatic steatosis induced by tributyltin (TBT, a fungicide with obesogenic properties), which increased white adipose tissue and adipocyte size and number [301]. Furthermore, recent works suggest a chromatin-remodeling effect of TBT beyond DNA methylation, which is conserved in mice and humans and can ultimately be reconstructed [302]. Noteworthy is the potential implications for human health derived from recent studies suggesting that humans might be susceptible to environmental disruptions of chromatin organization.

Besides the prominent effects of DNA methylation and transposable elements for reprogramming and transgenerational epigenetic inheritance, their impact on mammals suggests having noncoding RNAs (ncRNAs) as key mediator molecules. As a clear example, the locus *Rasgrf1* requires *Mili* and *Miwi2* (PIWI-interacting RNAs, piRNAs) for de novo promoter methylation in differentially methylated regions (DMR) in the male germline [303]. Therefore, the role of ncRNAs is becoming more evident as a mechanistic link between paternal environmental exposure and their offspring phenotype. In F0 mice, an early maternal separation causes the downregulation of the piRNA cluster 110 [304], while a high-fat and high-sugar diet induces the differential expression of 190 piRNAs distributed in 63 clusters [305], indicating that acquired stress and food-induced trait inheritance, at least, is enhanced by ncRNA-dependent regulation, and observation later verified as well in rats in a miRNA *let7c*-dependent manner [306].

From the different ncRNA biotypes, the regulation of transgenerational inheritance has been studied more extensively for miRNAs and tRNAs. It is now accepted that small ncRNAs can mediate non-Mendelian inheritance of traits acquired during life, and they are, in fact, abundant in mammals’ sperm [307]. The epigenetic modification of the mouse *Kit* gene regulation is a clear example of miRNA-mediated inheritance. The wild-type progeny of *Kit* heterozygous parents shows an altered *Kit* phenotype (whitetail) for consecutive generations by *miR-221-* and *miR-222*-direct targeting [308]. The exposure to microRNAs of the early embryonic genome seemed at the time convincing and promising to induce a permanent and heritable epigenetic change in gene expression.

Rodgers et al. demonstrated in 2015, through zygote microinjection, a nine-miRNA (*miR-29c*, *miR-30a*, *miR-30c*, *miR-32*, *miR-193-5p*, *miR-204*, *miR-375*, *miR-532-3p*, and *miR-698*) signature in sperm, related to paternal stress, which induces a reduction in the levels of mRNAs from the mother in zygotes at early stages, ultimately reprogramming gene expression in the offspring’s hypothalamic–pituitary–adrenal (HPA) stress axis, and recapitulating the phenotype of offspring stress dysregulation [309]. Of note, single miRNA injections did not affect the responsivity to the activated HPA axis in terms of corticosterone levels, which highlights the necessity of an orchestrated miRNA panel response. Of relevance for stress response and metabolic regulation, *miR-375* targets catenin B1 (*Ctnnb1*) in the mouse hippocampus. Notably, the reported miRNAs signature altered in mice induced rather systemic effects, and their expression was affected in serum and brain cortical regions [304].

Another example of small noncoding RNAs that are involved in transgenerational stress comes from 5′ halves of tRNAs, as found in the F3 generation sperm of rats that were pregnant during vinclozolin exposure [310]. Interestingly, another research group suggested that comparable behavioral, metabolic, and molecular effects were induced by either direct exposure to unexpected-maternal stress (MSUS) during early postnatal life or by injection of sperm small ncRNAs from MSUS males. In summary, cumulative evidence suggests that the environment itself can induce epigenetic transgenerational inheritance and, consequently, pollution-associated diseases, which indicates the relevance of further studies to evaluate its role as critical determinants influencing the etiologic, toxicologic, and progression factors of disease.

## 7. Future Perspective: Biomonitoring and Novel 3D Epigenomic Technologies

Although new drugs for several human cancer subtypes, presumably induced by epigenetic pollutants, are not yet approved at a clinical stage, the confluence of recent factors related to the development of sequencing technologies to analyze the human epigenome may change this scenario. Moreover, other factors such as an increased interest in research associated with environmental epigenetics, an improved understanding of the etiology and markers of cancer subtypes and their epigenetic landscape, and the promising results from immunotherapy might as well open new scopes.

The number of vulnerable individuals waiting for curative therapies is growing, in parallel to the growth in global industrial development. There are 1.9 million individuals living with pollution-associated diseases within the US alone, 3.9 million in Europe, and an estimated 19.3 million people worldwide, most of these cases due to cancer (2022 American Cancer Society) [311]. As it has been recently estimated, air pollution causes up to 3.1 million premature deaths worldwide every year, corresponding to 3.2% of the global disease burden [312]. Regionally, countries with low and middle incomes in the regions of Southeast Asia and Western Pacific had the most significant burden related to air pollution (2.2 and 2.8 million deaths, respectively). More than 287,000 deaths occurred in Europe, 131 000 in America, 600,000 in Africa, and 394,000 in the Eastern Mediterranean region [312]. On top of it, a currently unknown number of individuals harbor pre-symptomatic pathology, mainly children living around highly contaminated regions, which then represents an under-estimated population of individuals who may benefit from future treatments. A critical component for the approval of new therapies is selecting, validating, and deploying tools for disease screening and treatment monitoring. Here, we provide an overview of the ncRNA biomarkers acting as the driving force for pathogenesis. The development of sequencing technologies such as RNA-seq, small RNA-seq, single-cell RNA-seq (scRNA-seq), and nascent RNA-seq (GRO-seq, PRO-seq) significantly improves the integrative approaches for ncRNA research [313]. Recently developed techniques can be used to screen functional ncRNAs, such as global RNA interactions with DNA by deep sequencing (GRID-seq) to determine all potential chromatin-interacting RNAs [314]. Another important tool comes from public databases since they provide major references based on theoretical analysis, sequencing data, and even experimental verification, which guide the identification and the functional investigation of ncRNAs involved in human diseases [313].

Tremendous efforts are desired to discover and utilize ncRNAs as biomarkers in clinical diagnosis, calling for technological advancement in the analysis of circulating ncRNAs in biospecimens. Point-of-care testing based on microfluidic technology allows faster detection and diagnosis of diseases near the patient site than conventional lab-based testing [315]. It focuses on developing miniature laboratory-based procedures into user-friendly platforms that are portable, simple, and disposable [316]. This technology has been proposed for the integration for the extraction and purification of ncRNA from body fluids and in situ analysis. However, it is an ongoing challenge to integrate all these beneficial elements in a singular, convenient, and proof platform [316].

As technologies evolve, global interactions between ncRNA influencing chromatin modifications are still not covered. For example, lncRNAs exert their functionality by tuning chromatin architecture, resulting in the alteration in transcription readouts [317]. As Mishra and Kanduri referred, there are no unifying principles to understand the chromatin associations with lncRNAs [317]. Therefore, in the context of environmental exposures, the integrative mechanism that mediates the effect of ncRNA on chromatin remodeling processes is something that could be explored in the future to understand the effects of the exposome in a global manner.

Even though extensive research directed to characterize reprogrammable epigenetic signatures in pollution-associated diseases is still pending. Preparing the healthcare system for the advent of disease-modifying therapies against the effects of industrialization is imperative. It will be necessary to identify the most suitable early biomarkers to facilitate appropriate upcoming treatment, primarily for affected populations. We aimed to summarize recent developments in environmental biomonitoring and the validation of ncRNA signatures for pollution-associated pathologies. We discussed potential approaches that could be adopted to screen for and clarify the underlying pathology in people seeking medical advice because of late-stage symptoms. Future clinical trials should incorporate both CT scanning and radio-imaging and biomarker approaches by next-generation sequencing to assess the nuclear changes and the biological response at the same time. Given the potential barriers which may impede access to proper therapy in isolated communities located in high-exposure areas, the need to expand treatment options beyond specialized centers towards fluid biomarkers by non-invasive approaches is evident. While the proposed recommendations need empirical data for support, they represent a testable scenario regarding how upcoming clinical trials could be designed and treatments could be delivered in the clinic with the help of biomarkers. We briefly review recent data regarding ncRNA biomarkers for pollution-associated cancer in humans, highlighting the underlying epigenetic dysregulation in these patients. There is an evident need for further research into comorbidities induced by common environmental pollutants to suggest how different biomarkers could be used (most likely in combination as “disease epigenetic signatures”) to facilitate the development and clinical implementation of novel drug candidates against pollution-associated diseases.

## Figures and Tables

**Figure 1 biomolecules-12-00513-f001:**
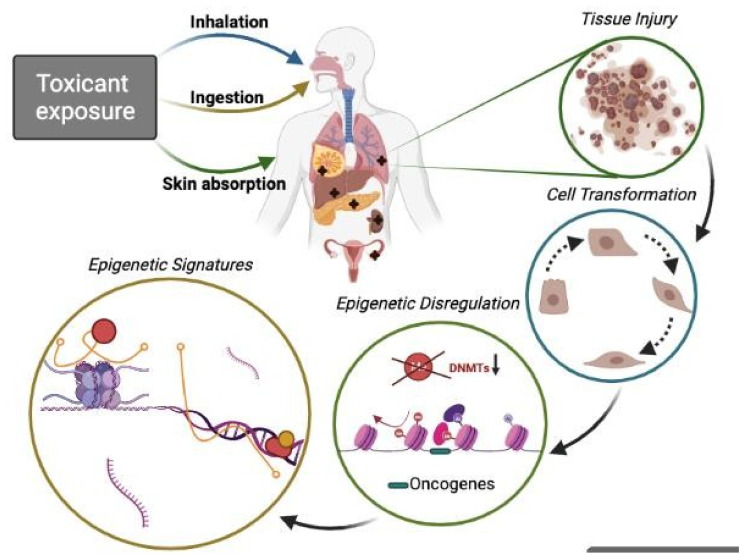
Epigenetic toxicants lead to the phenotypic transformation of normal cells. People exposed (by inhalation, food or water ingestion, and skin contact) to certain pollutants suffer potential tissue injury in the lung, mammary gland, liver, pancreas, skin, colon, ovary, and hematological tissue among other target organs. It is well known that acute or chronic exposures to toxicants are associated with malignant cell transformation and, thus, pollution-related diseases, including cancer. Aberrant genetic modifications (DNA methylation, histone modifications, and ncRNAs can transform cells and disturb the expression of genes involved in homeostasis maintenance. Recently much attention has been given to ncRNAs with their role in pathophysiological conditions and signaling pathways such as oncogenesis, cell survival, altered apoptosis, and cell adhesion. Thus, ncRNAs (miRNA, lncRNA) are vital mediator molecules. Moreover, other important regulators, such as ncRNA-associated proteins forming multi-component complexes on specific loci at specific time points that conform complex epigenetic signatures, are also crucial during the cell transformation process. Studying those epigenetic signatures may improve the understanding of the biology of different pollution-related cancer types.

**Figure 2 biomolecules-12-00513-f002:**
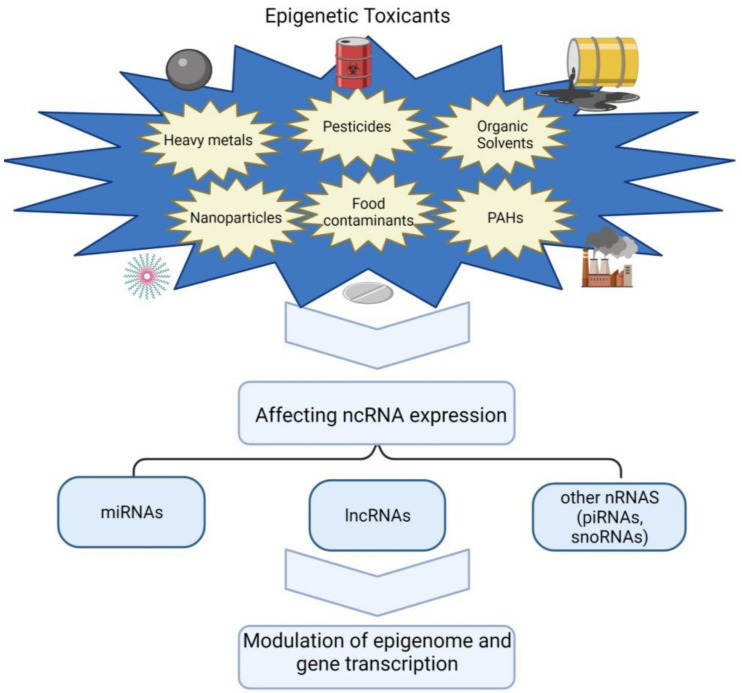
Classification of common and emerging epigenetic toxicants. The most common epigenetic pollutants are classified based on their chemical structure and associated epigenetic effects on different biological targets. Notably, the role of long noncoding RNAs (lncRNAs) in epigenetic processes has been recently highlighted as the primary mediator of the cellular response to environmental pollution.

**Figure 3 biomolecules-12-00513-f003:**
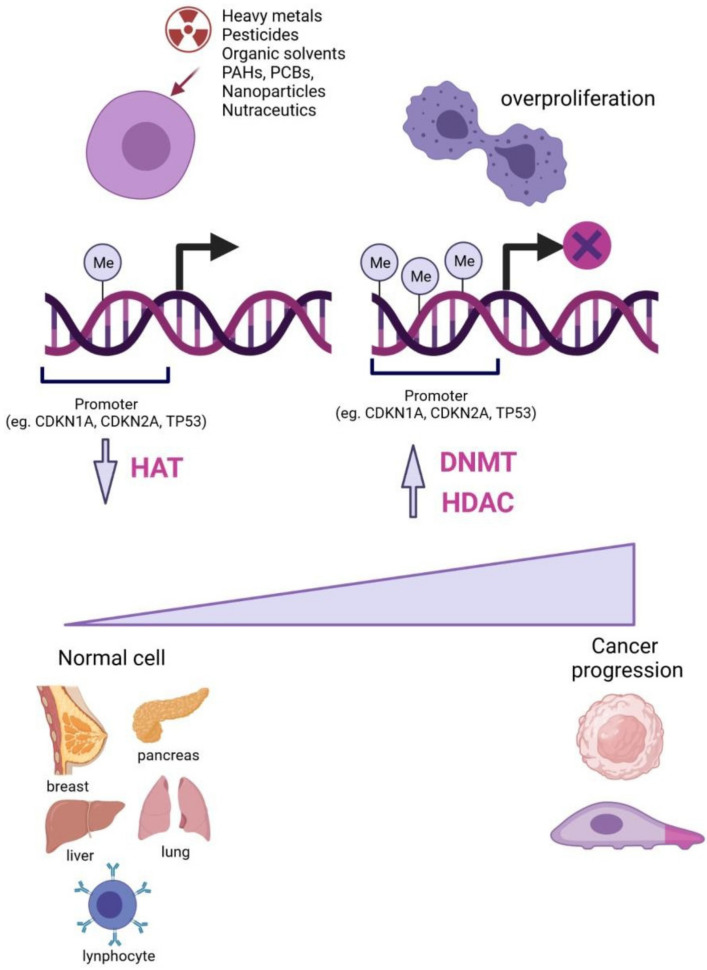
Epigenetic toxicants in humans induce chromatin structure changes. Exposure to toxicants is associated with hypermethylation in cell cycle regulators genes, causing hyperproliferation and cell transformation. PAHS: Polycyclic aromatic hydrocarbons; PCBs: polychlorinated biphenyl; HATs: histone acetyltransferases; DNMT: DNA methyltransferases; HDAC: histone deacetyltransferases.

## Data Availability

Not applicable.

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
