# Peer review of "Epigenetic Regulation in Exposome-Induced Tumorigenesis: Emerging Roles of ncRNAs"

_biomolecules, 2022, doi:10.3390/biom12040513_

Round 1

Reviewer 1 Report

In this review, the authors have summarized epigenetic regulation caused by toxicants and focused on non-coding RNA mediated epigenetic regulation. The authors have made in-depth reviews of non-coding RNAs, various types of toxicants and their effect on epigenetic alterations, and the dysregulation of non-coding RNAs in pollution-related cancer types. I only have very minor comments below:

The title is ‘Epigenetic regulation by ncRNAs in exposome-induced tumorigenesis’ but the authors have also included lots of epigenetic regulation that are not mediated by ncRNA alterations, like DNA methylation, which I found necessary and useful as well. I am just wondering if the title can be made to just ‘Epigenetic regulation in exposome-induced tumorigenesis’ and the authors can mention in the abstract that they will focus on the ncRNAs.

The authors mentioned that ‘They bind in a complementary manner in the 3′ untranslated region (3′ UTR) of messenger RNAs (mRNAs) to target them for degradation and block their translation’. Some miRNAs that have been reported to activate gene expression (for example, as reviewed in ‘Overview of MicroRNA Biogenesis, Mechanisms of Actions, and Circulation’). I would suggest the authors slightly modify this sentence to something like  ‘.... mostly target them for degradation and block their translation’, just to be more precise.

In the ‘Epigenetic alterations induced by toxicants’ section, I feel that some sub-sections don’t include citations/examples of epigenetics caused by toxicants, for example, ‘3.1.6. Particulate matter’ and ‘3.1.7. Indoor air pollutants’.

Author Response

Observations_Reviewer_1.

  1. In this review, the authors have summarized epigenetic regulation caused by toxicants and focused on non-coding RNA mediated epigenetic regulation. The authors have made in-depth reviews of non-coding RNAs, various types of toxicants and their effect on epigenetic alterations, and the dysregulation of non-coding RNAs in pollution-related cancer types. I only have very minor comments. The title is ‘Epigenetic regulation by ncRNAs in exposome-induced tumorigenesis’ but the authors have also included lots of epigenetic regulation that are not mediated by ncRNA alterations, like DNA methylation, which I found necessary and useful as well. I am just wondering if the title can be made to just ‘Epigenetic regulation in exposome-induced tumorigenesis’ and the authors can mention in the abstract that they will focus on the ncRNAs.

Answer: We appreciate this interesting remark. Accordingly, we have modified the title to “Epigenetic regulation in exposome-induced tumorigenesis: emerging roles of ncRNAs”.

  1. The authors mentioned that ‘They bind in a complementary manner in the 3′ untranslated region (3′ UTR) of messenger RNAs (mRNAs) to target them for degradation and block their translation’. Some miRNAs that have been reported to activate gene expression (for example, as reviewed in ‘Overview of MicroRNA Biogenesis, Mechanisms of Actions, and Circulation’). I would suggest the authors slightly modify this sentence to something like ‘.... mostly target them for degradation and block their translation’, just to be more precise.

Answer: We appreciate this comment, the reviewer is right on the pertinence to comment the existence of an alternative way of action for miRNAs. We have added a summary of this mechanism in Lines 91-97. Note: Line numbering along this letter makes reference to the revised version in Track-Changes mode, “All markup”.

“They bind in a complementary manner in the 3′ untranslated region (3′ UTR) of messenger RNAs (mRNAs) and mostly to target them for degradation and blocking of their translation [17]. A miRNA can bind numerous mRNAs to inhibit their translation. Therefore, miRNAs are considered crucial post-transcriptional regulators of gene expression [164]. However, under certain cellular contexts (for instance, cell cycle arrest) they are recruited with AGO2 and FXR1 to specific loci with AU-rich elements (AREs) to ultimately activate translation [18-20].”

  1. In the ‘Epigenetic alterations induced by toxicants’ section, I feel that some sub-sections don’t include citations/examples of epigenetics caused by toxicants, for example, ‘3.1.6. Particulate matter’ and ‘3.1.7. Indoor air pollutants’.

Answer: We appreciate this valuable observation that will help for improving our manuscript. Several paragraphs were written from sections 3.1.6 to 3.1.7.3 (included in Lines 343-360, 379-386, 410-418, 462-464, 477-488, 499-507):

-Lines 343-360

“Specifically, several studies have revealed the effects of PM2.5 on the deregulation of miRNAs such as miR-4516 and miR-32, whose up-regulation is observed in lung cancer cells. Moreover, the down-regulation of RPL37, a target of miR-4516, enhanced the expression of LC3B, a critical hallmark of autophagy [70]. Meanwhile, miR-32 might function as a tumor suppressor to repress EMT [71]. Conversely, PM2.5 affects the expression of miR-194-3p and miR-16 in human bronchial epithelial cells (CSE-HBEpiCs) treated with cigarette smoke extracts, as well as in human hepatocellular carcinoma (HCC) cells. miR-194-3p inhibition, in turn, enhances apoptosis [72] while miR-16 de-regulation could enhance metastasis and EMT features [73]. More recently, studies demonstrated that PM2.5 also transform normal cells by lncRNA alteration. For example, MEG3 (maternally expressed gene 3) over-expression promotes autophagy and apoptosis by increasing TP53 (a tumor suppressor) in HBE cells after treatment with arterial traffic ambient PM2.5 (TAPM2.5) and wood smoke PM2.5 (WSPM2.5) [74]. MEG3 is also up-regulated in CSE-treated 16HBE cells and COPD tissues, leading to cell proliferation impairment by down-regulation of miR-218 [75]. Another up-regulated lncRNA by PM2.5 is LOC101927514, which binds to STAT3 to raise an inflammatory state in 16HBE cells [76].”

-Lines 379-386

“In addition, ROS induced by PM2.5 exposure may promote the expression of some lncRNAs such as loc146880 [82], which expression was positively correlated with RCC2  [83] that encodes the protein RCC2/TD-60, required for cell cycle progression [84]. Furthermore, it was shown that RCC2 can bind and stimulate the effect of the lncRNA LCPAT1 (lung cancer progression-association transcript 1) on cell autophagy and EMT after exposure to PM2.5 and CSE (cigarette smoke extracts) in lung cancer cells [85]. Therefore, high loc146880 and LCPAT1 levels might be associated with lung cancer development [82,85,83].”

-Lines 410-418

“…as well as the expression of ncRNAs. For example, some breast tumor-related genes such as EPHB2 (Ephrin type-B receptor 2) and LONP1 (Lon Peptidase 1) present altered methylation under PAH and NO2 exposures [68]. On the other hand, chronic Po expo-sure in lung adenocarcinoma cells enhanced their cancer stem cell properties through a long non-coding loc107985872-notch1 signaling pathway which could be recapitulated in vivo [90-91]. Similarly, MALAT1 could interact with miR-204 which results in increased ZEB1 (an EMT-related transcription factor) function which in turn enhances the EMT and malignant transformation in lung bronchial epithelial cells after Po exposure [92].”

-Lines 462-464

“as it may occur in congenital heart disease (CHD), where several circRNAs may be dysregulated as shown in fetal heart rat samples exposed to formaldehyde compared to the control group [100]”.

-Lines 477-488

“Additionally, asbestos exposure is highly related to malignant pleural mesothelioma (MPM) development [103], in which various markers of DNA methylation, ncRNAs, and histone modifications have been proposed as diagnostic markers involved in the progression of MPM [104]. In this sense, lncRNA‐RP1 and miR‐2053 were part of a four-RNA signature in serum, in combination with the DNA damage regulated autophagy modulator 1 (DRAM1) and arylsulfatase A (ARSA) exclusively in MPM patients [105]. Furthermore, miR-16 (inhibitor of cell proliferation and migration), miR-17 (potential tumor suppressor), miR-126 (inhibitor of VEGF activity), and miR-486 (anti-fibrotic) were downregulated in MPM samples. Together, these miRNAs can represent the basis of the mechanism involved in MPM progression [106]. Since GAS5 (Growth arrest-specific transcript 5) might act as an oncogene released by the tumors and up-regulated in serum of MPM subjects, it is suggested as a potential circulating marker [107].”

-Lines 499-507

“A recent study proposed the long non-coding NEAT1 and a panel of four-miRNAs (miR-301a-3p, miR-16-5p, miR-15b-5p and miR-15a-5p) as an efficient epigenomic sig-nature related with toluene-induced neurodegeneration [109]. Conversely, another study showed that the effects of chronic combined exposure to volatile organic compounds (VOCs, toluene, ethylbenzene, and xylene) are more unfavorable than those resulting from a single compound exposure [84]. Specifically, MALAT1 and eight coding genes (CACNG8, CLIP2, CNTNAP3, FMR1, GLS2, SULT4A1, TP73, and WNT7B) expressions were significantly reduced by DNA hypermethylation in VOC-exposed subjects [110].”

Reviewer 2 Report

In this manuscript, the authors focus on the role of environmental factors on diseases’ onset and progression. Specifically, they highlight their role in DNA modifications and ncRNA regulation.

The argument is interesting and the review updated, however, in the present form, the manuscript suffers of some criticisms that should be addressed.

Major points:

  • The ncRNA-based therapeutic approaches seems to represent a good strategy in disease progression. I suggest to extend the discussion in a more extensive manner (for refs see also Winkle 2021 nat rev drug disc, Battistelli 2021 cancer res).
  • The effects of some of the presented environmental factors on epigenetic landscape and chromatin modifications (histone modifications) should be more extensively discussed.
  • The balance between advantages and toxic effects of nanomaterials/nanoparticles should be discussed on the basis of literature data.

Minor points:

Some typing errors should be amended.

Author Response

Observations_Reviewer_2.

  1. In this manuscript, the authors focus on the role of environmental factors on diseases’ onset and progression. Specifically, they highlight their role in DNA modifications and ncRNA regulation. The argument is interesting and the review updated, however, in the present form, the manuscript suffers of some criticisms that should be addressed. The ncRNA-based therapeutic approaches seems to represent a good strategy in disease progression. I suggest to extend the discussion in a more extensive manner (for refs see also Winkle 2021 nat rev drug disc, Battistelli 2021 cancer res).

Answer: We appreciate this valuable observation that will help for improving our manuscript. A completely new section has been added as “5.10. Potential role of ncRNAs as therapeutic agents in exposome-associated diseases” ( Included in Lines 1383-1425). Note: Line numbering along this letter makes reference to the revised version in Track-Changes mode, “All markup”.

“5.10. Potential role of ncRNAs as therapeutic agents in exposome-associated diseases

Since ncRNAs have a functional role in modulating modulate the expression of multiple downstream gene targets and associated pathways in cancer and other diseases, there is a rationale that suggests that ncRNAs-based therapies are promising as therapeutic options [176]. Different ncRNA-based strategies have been proposed for therapeutical purposes: small interfering RNAs (siRNAs), antisense oligonucleotides (ASOs), short hairpin RNAs (shRNAs), anti-microRNAs (anti-miRNAs), miRNA mimics, miRNA sponges, therapeutic circular RNAs (cirRNAs), and CRISPR-Cas9-based gene editing [281]. Several clinical trials have confirmed that ncRNAs are successful for the treatment of complex diseases including cancer [282,176]. According to a recent review by Winkler et al., 11 therapeutics based on ncRNAs have been approved by the FDA and/or the European Medicines Agency (EMA), and many other ncRNA approaches are under phase  II or III clinical testing. The majority of the approved therapeutics correspond to siRNAs or ASOs that promote downregulation of genes, or that target pre-mRNA splicing. It is important to remark that there are no ncRNA-based strategies that have entered clinical testing so far [215]. However, recent investigations highlight their potential as ncRNA-based therapeutics. For example, Battistelli et al. studied a mutant form of the ncRNA HOTAIR to reduce the epithelial-to-mesenchymal transition (EMT). Their results showed that the HOTAIR mutant form was able to reduce the binding of HOTAIR to Snail, impairing EMT [283].

Although ncRNAs represent a novel therapeutical option, there are still some limitations to their adoption in clinical practice [184]. Trials have reported ambivalent results with some studies reporting positive effects whereas others showed limited success or inducing toxicity [185]. Issues related to specificity ( undesired on-target effects), delivery, and tolerability hamper their translation into the clinical scenario [185,184]. However, novel approaches to overcome these limitations are under investigation such as chemical modification of the backbone of the nucleic acid sequence [284], and the use of nanoparticles for improving their delivery and stability as biocompatible NP carries protects against degradation by nucleases [285,146].

Several technological advances have increased the capacity to design and manufacture most ncRNA mimics. However, a clear limitation for their use is the current lack of deep  understanding of their functional roles in oncogenesis and their downstream genetic targets [286]. For example, the therapeutic application of transfer RNAs, circular RNAs, small nucleolar RNAs, and piwi-interacting RNAs is still at its infancy [286].”

  1. The effects of some of the presented environmental factors on epigenetic landscape and chromatin modifications (histone modifications) should be more extensively discussed.

Answer: We appreciate this valuable observation, three paragraphs were written to include this suggestion along our manuscript:

Paragraph 1 (1. Introduction section, Lines 50-58)

“The adaptation of cells to environmental factors causing stress relies on a wide range of tightly controlled regulatory mechanisms. The epigenetic landscape represents the platform where multiple environmental factors interact with the complex genetic milieu, resulting in alterations in the expression gene that shape many aspects of health and disease [7]. Changes in the organization and the structure of chromatin structure are associated with the transcriptional response to stress caused by the environment, and in some cases, can impart the memory of stress exposure to subsequent generations through mechanisms of epigenetic inheritance [8].”

Paragraph 2 ( 3.4.1 Nanotecnotechnology section, lines 777-785)

“It is important to highlight that besides ncRNAs,  histone post-translational modifications (PTMs), including methylation and acetylation, are covalent modifications that have an essential role in gene regulatory function by modulating chromatin structure [45,8]. Generally, the repressive state of chromatin is enriched in H3K9 and H3K27 trimethylation (H3K9me3 and H3K27me3), while H3K4me3 is enriched at promoter regions and associated with more accessible chromatin [160]. Besides methylation and acetylation, other modifications such as phosphorylation of histones, are indicative of toxicity-induced damage or environmental stress [161]. In the context of nanotechnology products,…”

Paragraph 3 ( Future perspective section, lines 1611-1619)

“As technologies evolve, global interactions between ncRNA influencing chromatin modifications are still not covered. For example, lncRNAs exert their functionality by tuning chromatin architecture, resulting in alteration in transcription readouts [317]. As Mishra and Kanduri referred, there are no unifying principles to understand the chromatin associations with lncRNAs [317]. Therefore, in the context of environmental exposures, the integrative mechanism that mediates the effect of ncRNA on chromatin remodeling processes is something that could be explored in the future to understand the effects of the expo-some in a global manner.”

  1. The balance between the beneficial and detrimental effects of nanomaterials/nanoparticles should be discussed based on literature data.

Answer: We appreciate this valuable observation that will help for improving our manuscript. A completely new section has been added as “3.4.1.4. The balance between the beneficial and detrimental effects of nanomaterials and nanoparticles” ( Included in Lines 848-890):

“The main goal of nanotechnology is to develop smart nanomaterials that improve life quality without inducing adverse effects [172]. This fact explains why their wide usage and the global market are increasing in an exponential manner [173]. There are many pros and cons for their adoption; one of the most promising applications of NMs is in the field of medicine and pharmacology. For example, nanoparticles and nanomaterials have been designed as drug delivery systems and therapeutics. Besides their potential application as antitumor drugs [174], they can be used as carriers to deliver proteins, vaccine antigens, and drugs to a specific site of action [175]. A particular application in the context of ncRNA-based therapy (discussed in section 5.10) is that nanoparticles have been developed for the safe transference of nuclei acids and increasing their stability to reach their targets [176,177]. Moreover, the application of NMs for the improvement of the quality environment is promising for the reduction of contaminants. NMs such as photocatalyst, nanosized adsorbents, nanomembranes, have been developed as emerging technologies for water purification [178]. 

On the other hand, NMs may have a double-sword effect. Their widely used ap-plication should be taken with caution as the exponential growth of nanotechnology usage imposes concerns over its impact on human and environmental health and safety (EHS) [173]. It is important to recognize that information about their environmental fate, transformation, transport, and accumulation in other environmental compartments (e.g. air, air, soil, and water) are still under study. Attention has been focused on their accumulation can be selective by aquatic environments [179]. Furthermore, there is evidence indicating that many NPs possess DNA damaging activity; however, the conclusions are still controversial [173]. As epigenetic mechanisms have an important role in the onset and progression of diseases, some attention should be given to get more insides correlation between epigenetic alterations and diseases, specifically for understanding the pathogenesis associated with nanomaterials. There are still some questions regarding the nano-size effect of materials ( optical, thermal, magnetic, and the induction of reactive species, in the regulation of the epigenetic landscape, remain open. Taking both pros and cons of nanoparticles and nanomaterials, we consider that more research should be performed in the field of nanotoxicology to provide indications regarding their widespread usage in the future. As epigenetics represents a novel endpoint in the field of nanotoxicology [172], it should be important to increase the evidence for epigenetic toxicity in human-based biological models to distinguish between adverse health effects of NP exposure, in contrast to adaptative mechanisms to these nanoparticles.  Possibly in the future, novel NMs could be developed to counterattack the effects of other NMs.”

  1. Some typing errors should be amended.

Answer: We appreciate this observation and apologize for this inconvenience. We have corrected several misspelling and typing errors along the text in Track-changes mode.
